# FedCVD: A Federated Learning Benchmark for ECG Classification and Echocardiogram Segmentation

## Abstract

Cardiovascular diseases (CVDs) are currently the leading cause of death worldwide, highlighting the critical need for early diagnosis and treatment. Machine learning (ML) methods can help diagnose CVDs early, but their performance relies on access to substantial data with high quality. However, the sensitive nature of healthcare data often restricts individual clinical institutions from sharing data to train sufficiently generalized and unbiased ML models. Federated Learning (FL) is an emerging approach, which offers a promising solution by enabling collaborative model training across multiple participants without compromising the privacy of the individual data owners. However, to the best of our knowledge, there has been limited prior research applying FL to the cardiovascular disease domain. Moreover, existing FL benchmarks and datasets are typically simulated and may fall short of replicating the complexity of natural heterogeneity found in realistic datasets that challenges current FL algorithms. To address these gaps, this paper presents the first real-world FL benchmark for cardiovascular disease detection, named FedCVD. This benchmark comprises two major tasks: electrocardiogram (ECG) classification and echocardiogram (ECHO) segmentation, based on naturally scattered publicly available datasets constructed from the CVD data of seven institutions. Our extensive experiments on these datasets reveal that FL faces new challenges with real-world non-IID and long-tail data.

## 1 Introduction

Cardiovascular Diseases (CVDs) cause over 18 million deaths globally each year, positioning them as one of the most significant global health challenges (Donkada et al., 2023). Early detection and diagnosis of CVDs are crucial, as they allow for timely medical interventions and more effective treatment plans, which in turn significantly lower patient mortality rates (Aversano et al., 2022). Recently, with the growing availability of electronic health records and other high-quality clinical data, researchers have increasingly utilized machine learning techniques to automate clinical diagnostics (Yan et al., 2020; Chen, 2020), a strategy that has proven highly effective in CVDs (Alizadehsani et al., 2019; Al'Aref et al., 2019). This data-driven approach can facilitate efficient early screening and optimize the allocation of healthcare resources, improving overall patient outcomes.

However, medical studies usually face the issue of bias, that is, the data distribution is restricted by factors such as geography, and may even lead to discriminatory outputs. Therefore, multi-center collaboration is required, as it enables the utilization of richer regional and demographic characteristics, fostering more precise and comprehensive research outcomes. However, medical data is considered highly sensitive, and recent privacy regulations (e.g., EU General Data Protection Regulation (GDPR) (noa, 2016)) restrict its transfer, hindering the expansion of datasets through data sharing among institutions to train more efficient models, i.e., data isolation.

To address this issue, federated learning (FL) (Yang et al., 2019; McMahan et al., 2017) has been proposed as a more secure paradigm of distributed machine learning. A typical FL architecture involves a coordinator (Server) and several participants (Clients) with private data. By aggregating (e.g., FedAvg (McMahan et al., 2017)) the model parameters or gradients from different Clients on the Server, participants collaboratively train high-performance models keeping private data within

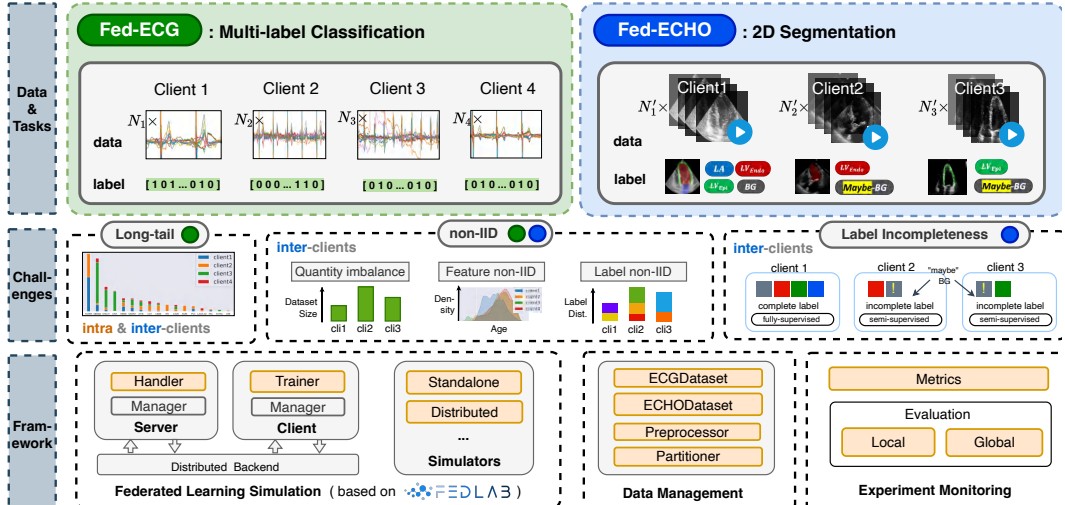

Figure 1: The overall architecture of the proposed FedCVD benchmark. We present two main settings (Fed-ECG, Fed-ECHO) and an experimental platform, highlighting three primary challenges. Green and blue circles in the challenges section indicate their presence in Fed-ECG and Fed-ECHO, respectively. The API section highlights user-facing APIs in orange boxes.

their respective domains. This process only involves transmitting model parameters, thus ensuring a certain degree of data privacy.

In medical applications such as CVD, integrating FL enables medical institutions to harness larger and more diverse datasets, collaboratively training models that are more unbiased and generalized, thereby enhancing diagnostic accuracy and clinical decision-making in real-world settings. For instance, Yang et al. (2021) applied FL for joint case analysis across three institutions during the Covid-19 pandemic, significantly improving CT segmentation performance and facilitating more accurate detection of Covid-19. Additionally, the effectiveness of FL has been demonstrated in multi-center research, such as a study involving three centers focused on the medical image analysis task of whole prostate segmentation (Sarma et al., 2021), further underscoring its relevance in realistic, large-scale medical scenarios.

Facilitating the application of FL in multi-center medical research, particularly in areas such as CVD, necessitates the creation of appropriate datasets and benchmarks to support the development of robust algorithms. However, publicly available cardiovascular disease datasets are limited, and those that do exist often suffer from incompatibility due to variations in data collection protocols. Furthermore, there is currently no comprehensive, publicly accessible benchmark specifically designed for FL on CVD data, which significantly impedes research progress in this domain. Additionally, most existing FL benchmarks simulate an FL scenario by manually partitioning data—often without considering geographic distribution—into smaller subsets, resulting in an overly idealized model that fails to capture the complexities and heterogeneity of real-world, multi-center CVD scenarios. This gap presents substantial challenges for the development and validation of effective FL algorithms in practical medical applications.

To address these gaps, we introduce the *first* multi-center FL benchmark specifically designed for CVD tasks, named **FedCVD**. Built from real-world CVD data from seven medical institutions (i.e., clients, the two terms will be used interchangeably), FedCVD utilizes a *natural partitioning* strategy. It comprises two primary datasets along with their corresponding tasks: electrocardiogram (ECG) classification and echocardiogram (ECHO) segmentation. FedCVD encapsulates three critical traits of FL in real-world CVD applications, each of which presents substantial challenges to FL algorithms:

**Challenging Trait 1. Non-IID Data**: The Non-independently and identically distributed (non-IID) data among institutions, including non-IID feature (e.g., variations in imaging quality due to different equipment across institutions) and non-IID label (e.g., differences in disease prevalence across regions). The non-IID data may significantly hindering global model convergence.

Table 1: Comparison of FedCVD with Other Federated Datasets or Benchmarks.

| | Long-tailedness Considered | Natural Partition | Incomplete Label | Covers CVD Domain | Code Available |
|---|---|---|---|---|---|
| FedDTI (Mittone et al., 2023) | ✗ | ✗ | ✗ | ✗ | ✓ |
| FedTD (Lindskog & Prehofer, 2023) | ✗ | ✗ | ✗ | ✗ | ✗ |
| Flamby (du Terrail et al., 2022) | ✗ | ✓ | ✗ | ✗ | ✓ |
| NIPD (Yin et al., 2023) | ✗ | ✓ | ✗ | ✗ | ✓ |
| FEDLEGAL (Zhang et al., 2023c) | ✓ | ✓ | ✗ | ✗ | ✓ |
| FLHCD (Goto et al., 2022) | ✗ | ✓ | ✗ | ✓ | ✓ |
| FedMultimodal (Feng et al., 2023) | ✗ | * | ✗ | ✗ | ✓ |
| FedAudio (Zhang et al., 2023b) | ✗ | * | ✗ | ✗ | ✓ |
| FedCVD | ✓ | ✓ | ✓ | ✓ | ✓ |

*: Some datasets included are naturally partitioned.

**Challenging Trait 2. Long-tail Distribution**: The labels of CVD data from various institutions exhibit a long-tailed distribution, where a few labels dominate while most labels are sparse. This challenges the model's performance on tail classes, a problem that is exacerbated in FL scenarios.

**Challenging Trait 3. Label Incompleteness**: For the same type of medical images, hospitals with strong annotation capabilities can identify all key segmentation areas, while those with weaker capabilities can identify only some. This incomplete annotation can mislead the global model's segmentation performance in areas unrecognized by certain institutions.

Focusing on these challenging traits, FedCVD provides new insights and evaluation metrics for designing FL algorithms in multi-center CVD scenarios. Our contributions are summarized as follows:

1. We introduce FedCVD, an open-source federated multi-center healthcare dataset and benchmark specifically for the CVD domain. To the best of our knowledge, FedCVD is the largest multi-center CVD benchmark available. This dataset encompasses two critical tasks—multi-label classification and segmentation—within the CVD domain and includes data of varying scales. Crucially, all datasets are partitioned using natural splits.

2. Our benchmark emphasizes three critical traits in the FL-CVD scenario: non-IID, long tail, and label incompleteness. These traits pose significant challenges to existing FL algorithms.

3. We conducted extensive experiments on FedCVD to evaluate the performance of mainstream FL and centralized learning methods, validating the effectiveness of FL in the CVD context and the proposed three challenges. Additionally, we have made the open-source code available at https://anonymous.4open.science/r/ZYNTMBB-8848, ensuring benchmark reproducibility and facilitating seamless integration into various FL frameworks.

## 2 RELATED WORK

**AI for CVD Research.** Numerous studies have leveraged CVD data for disease detection and diagnostic support, focusing primarily on ECG and ECHO data. ECG, recorded as time-series signals, captures the heart's electrical activity and provides insights into cardiac conditions and potential damage (Donkada et al., 2023). These studies often frame ECG-based tasks as classification problems for disease diagnosis and heart metric analysis (Thanapatay et al., 2010; Muirhead & Puff, 2004; Christov et al., 2005; Behadada & Chikh, 2013; Zhang et al., 2021). ECHO, comprising ultrasound images, enables real-time visualization of heart chambers and blood flow, aiding in diagnoses of conditions like heart valve disorders and Congestive Heart Failure (CHF). For instance, Goto et al. (2022) used ECHO data for Hypertrophic Cardiomyopathy detection, while Ostvik et al. (2019) applied Convolutional Neural Networks (CNNs) for standard view classification to enhance clinical efficiency. Automated ECHO segmentation, crucial for assessing heart morphology and diagnosing conditions such as myocardial infarction, addresses the limitations of manual segmentation, which is time-consuming and subjective. Studies have employed AI models for ventricular (Zhang et al., 2014; De Alexandria et al., 2014; Qin et al., 2013; Kiranyaz et al., 2020b) and atrial segmentation (Haak et al., 2015b;a). Although these works highlight the potential of AI in analyzing CVD data, they are predominantly restricted to single-institution settings.

**FL for Multi-center CVD Research.** CVD research necessitates multi-center collaboration, and FL presents a promising solution. Most current studies simulate FL in multi-institution collaborative training by manually partitioning data from a single institution. For instance, Sakib et al. (2021) trained a classification model to detect cardiac arrhythmia using ECG data within a federated architecture, partitioning data from the MIT-BIH Supraventricular Arrhythmia database (Greenwald et al., 1990). Similarly, Zou et al. (2023) investigated congestive heart failure detection in a federated setting by splitting samples from the NSR-RR-interval and CHF-RR-interval databases (Goldberger et al., 2000) into 2 to 4 clients for simulated training. FedCluster (Lin et al., 2022) tackled the issue of unbalanced class distributions in ECG data by optimizing algorithms that cluster local parameters before performing intra- and inter-class aggregation, thus increasing the weight of minority classes. Their data were also partitioned from the MIT-BIH Arrhythmia database (Goldberger et al., 2000). However, these partition-based simulations may not fully capture the true distribution characteristics of CVDs. In contrast, FLHCD (Goto et al., 2022) demonstrated federated training for hypertrophic cardiomyopathy detection using ECG and ECHO data from four medical institutions (three in the US and one in Japan), showcasing the effectiveness of FL in a naturally partitioned, multi-center setting.

**FL Benchmarks.** To further support research in FL, numerous datasets and benchmarks have been proposed for a wide range of applications. A comprehensive comparison of FedCVD with these benchmarks is shown in Table 1. Existing studies often manually partition centralized datasets and introduce perturbations or masking to features and labels to mimic the heterogeneity found in real-world FL scenarios. For instance, FedTD (Lindskog & Prehofer, 2023) and FedDTI (Mittone et al., 2023) simulate non-iid data partitioning by altering feature distributions and sample sizes.

Since manual data is incapable of capturing real-world challenges, real-world multi-institution benchmarks are essential. Several FL benchmarks directly utilize real-world multi-institution data. For instance, NIPD (Yin et al., 2023) employs data from cameras in different geographical locations as FL clients for person detection tasks, naturally exhibiting non-iid characteristics. Similarly, FEDLEGAL (Zhang et al., 2023c) provides a FL benchmark for NLP tasks in the legal domain, using geographically distributed case-based text data for natural data partitioning. Another example is FLamby (du Terrail et al., 2022), an FL benchmark for real-world distributed medical data, offering seven datasets naturally distributed by geography or institution, with corresponding tasks including segmentation and binary/multiclass classification for medical image analysis and diagnostic assistance. Some benchmarks combine natural partitioning with simulated partitioning. For instance, FedAudio (Zhang et al., 2023b) applies simulated partitioning for certain audio data, while introducing perturbations to mimic noisy data and labels. FedMultimodal (Feng et al., 2023) uses a Dirichlet distribution to partition multimodal data from various domains, incorporating missing modalities, labels, and erroneous labels to simulate real-world heterogeneity. Despite these advances, none of these benchmarks cover the CVD domain. Although FLHCD (Goto et al., 2022), which utilizes multi-institution data for hypertrophic cardiomyopathy detection, has a setup most similar to ours, it does not address challenges such as the long-tail distribution and incomplete label issues, which are specifically tackled by FedCVD.

## 3 THE PROPOSED FEDCVD

In this section, we present the details of the proposed general FL framework for healthcare tasks as shown in Figure 1. Our framework is built upon the lightweight open-source framework FedLab (Zeng et al., 2023) for FL simulation. We present the details of datasets, metrics, and baseline models in Section 3.1. Then, we discuss the main FL challenges that FedCVD supported in Section 3.2.

### 3.1 DATASETS

Figure 1 provides an overview of the datasets included in FedCVD. In this section, we provide a brief description of each dataset.

**Fed-ECG.** The 12-lead ECG signals in Fed-ECG are sourced from four distinct datasets. The first and third datasets were collected from Shandong Provincial Hospital (Liu et al., 2022) and Shaoxing People's Hospital (Zheng et al., 2022) in China, respectively. The second dataset is from the PTB-XL database, released by Physikalisch Technische Bundesanstalt (PTB) (Wagner et al., 2020),

Table 2: Overview of the datasets, tasks and metrics in FedCVD.

| Dataset | Fed-ECG | | | | Fed-ECHO | | |
|---|---|---|---|---|---|---|---|
| Task Type | Multi-label Classification | | | | 2D Segmentation | | |
| Input | 12-lead ECG Signal | | | | Echocardiogram | | |
| Prediction (y) | Diagnostic Statement | | | | Cardiac Structure Mask | | |
| Data source | SPH | PTB-XL | SXPH | G12EC | CAMUS | ECHONET-DYNAMIC | HMC-QU |
| Preprocessing | Label Alignment | | | | Resizing and Label Alignment | | |
| Patient Size | 21,530 | 16,699 | 36,272 | UNKNOWN | 500 | 10,024 | 109 |
| Sample Size | 22,425 | 19,019 | 36,272 | 6,205 | 1000 | 20,048 | 2,349 |
| Metrics | Micro F1 / mAP | | | | DICE / Hausdorff distance | | |
| Input Dimension | $12 \times 5000$ | | | | $112 \times 112$ | | |

and the fourth originates from the PhysioNet/Computing in Cardiology Challenge 2020 (Alday et al., 2020), which represents a large population from the Southeastern United States. These four datasets, originating from geographically diverse regions, are naturally suited for the FL setting due to their separation by location.

The original four datasets consist of ECG data with varying lengths and labels, each based on different standards, such as AHA (Kligfield et al., 2007), SCP-ECG, and SNOMED-CT, making them incompatible for use in a FL setting directly. To standardize ECG lengths, we truncate signals longer than 5000 samples and apply edge padding to those shorter than 5000. Additionally, we retain only samples whose labels appear in at least two datasets, ensuring alignment across labels. Figures 2(a) and 2(b) illustrate the heterogeneity in both age and label distributions among institutions. Appendix D provides further details on the dataset and the preprocessing pipeline.

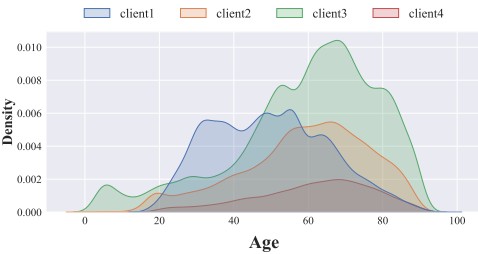

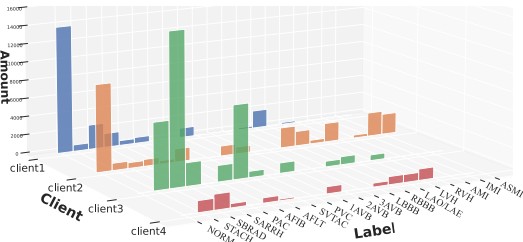

(a) Feature non-IID of the Fed-ECG dataset, demonstrated with the non-identical distribution of patient age among institutions.

(b) Label non-IID of the Fed-ECG dataset, shown as the variation in the number of each label (right axis) across different institutions (left axis).

Figure 2: Demonstration of the non-IID nature of Fed-ECG Dataset.

**Fed-ECHO.** This dataset is derived from three sources: CAMUS (Leclerc et al., 2019), ECHO-DYNAMIC (Ouyang et al., 2020), and HMC-QU (Kiranyaz et al., 2020a). CAMUS provides a database of gray-scale 2D apical four-chamber echocardiographic images, acquired at the University Hospital of St. Etienne in France, fully annotated with the left ventricular endocardium ($LV_{Endo}$), epicardium ($LV_{Epi}$), and left atrial wall (LA) regions. ECHO-DYNAMIC contains 2D echocardiogram videos collected at Stanford Medicine, with only the $LV_{Endo}$ region annotated in two frames. HMC-QU, released through a collaboration between Qatar University (QU) and Hamad Medical Corporation (HMC) Hospital, includes 2D echocardiogram videos from Qatar, with annotations limited to the $LV_{Epi}$ region in frames of a single cardiac cycle.

For consistency among each institution, we only select annotated frames for the experiment. The followed image pre-processing pipeline includes picture resizing to $1 \times 112 \times 112$, and label alignment. More details about this dataset are available in Appendix E.

### 3.2 CHALLENGING TRAITS OF FEDCVD

**Non-IID.** Non-independently and identically distributed (non-IID) is a typical characteristic in FL scenarios, encompassing non-IID features and non-IID labels, where clients' data shows heterogeneity

in both feature and label spaces. Quantity imbalance, where institutions hold uneven sample sizes, can further exacerbate these non-IID issues. Among these, non-IID labels have the most pronounced impact on FL model performance. This is because the quantity and types of labels held by each institution can vary greatly, misleading the local supervised training process and causing "Client Drift" (Karimireddy et al., 2020), which hinders global model convergence.

Fed-ECG naturally exhibits these three characteristics. In terms of feature distribution, Figure 2(a) shows significant age distribution differences among institutions' patients, with Institution 1 notably younger, reflected in the ECG features. Regarding sample size, Figure 2(b) depicts significant differences among the four institutions, with Institution 4 having the fewest samples. For label distribution in the Fed-ECG multi-label classification task, each sample may belong to multiple categories, but the quantity and proportion of different labels vary significantly among institutions. For example, the most common label for Institution 1 and Institution 2 is NORM (Normal), while for Institution 3 and Institution 4 it is STACH (Sinus tachycardia). Some institutions may even lack samples with certain labels, such as both Institution 3 and Institution 4 lacking samples labeled as PAC (Atrial premature complex(es)). These non-IID characteristics challenge the four institutions in collaboratively training a multi-label prediction model, as institutions struggle to capture information about the labels they lack during local training, potentially leading to client drift.

**Long-tail Distribution.**   In addition to the inter-institution heterogeneity caused by non-IID labels, Fed-ECG also exhibits intra-institution and inter-institution heterogeneity in the form of a long-tail distribution of labels. Figure 2(b) illustrates a clear long-tail characteristic of each institution's internal label distribution, with a few dominant labels having many samples and numerous labels having fewer samples (long-tail). These tail categories are already troublesome during independent local training, as the model may neglect the tail categories. In FL scenarios with quantity imbalance and non-IID labels, the long-tail problem is further exacerbated. For instance, categories mainly found in the disadvantaged institutions' tails may be in an even worse position within the overall data of all institutions. The long-tail characteristic challenges FL algorithms in ensuring the effectiveness and fairness of handling samples from various categories across institutions.

**Label Incompleteness.**   Fed-ECHO presents the most challenging scenario: label-incomplete FL. In Fed-ECHO, three naturally formed institutions hold ECHO video data with annotations (image region segmentation). However, due to varying annotation capabilities, the completeness of labels among the three institutions differs, as shown in Figure 1. Institution 1 has the most complete labels (four labels) due to its ability to identify and annotate all four key regions (including the background). In contrast, Institution 2 and Institution 3 each have labels for only one key region ($LV_{Endo}$ and $LV_{Epi}$, respectively). This incompleteness introduces (1) label heterogeneity, similar to the label-non-IID in Fed-ECG, where Institution 2 and Institution 3 lack some labels, and (2) mislabeling, where Institution 2 and Institution 3 label unrecognized parts as background, conflicting with Institution 1's labels and causing misleading information. This scenario significantly challenges FL algorithms to effectively utilize the different levels of label completeness from each Institution and leverage highly heterogeneous data to benefit the global model.

## 3.3 TASKS & METRICS

**Fed-ECG.**   The corresponding task on Fed-ECG's four datasets involves multi-label classification for each institution, a challenging problem due to the large number of labels and the long-tail distribution inherent to the data. To provide a more fine-grained evaluation, we focus on detailed label distinctions, which are of particular interest to clinicians, rather than broader label categories. To thoroughly assess performance, we adopt two metrics: *micro-F1*, which evaluates the overall performance across all labels, and *mean average precision (mAP)*, which specifically measures the impact of the long-tail distribution on model performance.

To further illustrate the *long-tail distribution* challenge posed by Fed-ECG, we introduced two additional metrics, namely *F1-STD* and *Top-K*. The F1-STD metric measures the standard deviation of F1 scores across classes, reflecting the learning algorithm's ability to manage long-tail problems; the larger the F1-STD, the poorer the algorithm's performance in this regard. Top-K, on the other hand, refers to selecting the K classes with the most samples and the K classes with the fewest samples,

calculating the average F1 score for each group, and then computing the relative performance drop between them. A larger performance drop indicates a more severe long-tail problem.

**Fed-ECHO.** The common task across Fed-ECHO's three datasets is the automatic segmentation of cardiac structures in echocardiograms, a crucial step in further diagnosing cardiovascular diseases. This task is particularly challenging due to the varying quality of the original echocardiograms across datasets. To evaluate segmentation accuracy, we use both the *Dice similarity index (DICE)* and the *2D Hausdorff distance ($d_H$)*. The Dice index measures the overlap between the predicted segmentation and the ground truth, while the $d_H$ quantifies the local maximum distance between the two areas.

# 4 EXPERIMENT

## 4.1 EXPERIMENT DETAILS

**Baseline Algorithms.** Our experiments utilize seven typical FL algorithms across both datasets. The first four are classical global FL algorithms: *FedAvg* (McMahan et al., 2017), the oft-cited FL algorithm, collaboratively trains a global model across participants. *FedProx* (Li et al., 2020) addresses statistical heterogeneity in FL by introducing an L2 proximal term during local training, while *Scaffold* (Karimireddy et al., 2020) mitigates client drift through control variates and server-side learning rate adjustments. *FedInit* (Sun et al., 2023) also tackles client drift by employing a personalized, relaxed initialization at the start of each local training stage. The last three are personalized FL methods: *Ditto* (Li et al., 2021), which excels in balancing accuracy, fairness, and robustness in FL; *FedSM* (Xu et al., 2022), which combines model selection with personalized methods to avoid client drift; and *FedALA* (Zhang et al., 2023a), which reduces the impact of statistical heterogeneity by adaptively aggregating both the global and local models. For the Fed-ECHO dataset, we further evaluate two Federated Semi-Supervised Learning (FSSL) methods: *Fed-Consist* (Yang et al., 2021), which uses a consistency-based method for segmentation, and *FedPSL* (Dong et al., 2023), which applies separate model aggregation and meta-learning techniques for classification. In addition to the FL family, we include two other baseline algorithms: *Client*, which refers to training models using only local data without collaboration among participants, and *Central.*, which represents the ideal centralized training scenario where the server has access to all participants' data.

**Setup.** We took into account the models that are widely used within the field as the default implementations of the models in the experiments. For Fed-ECG, a residual network, with its implementation following that of Strodthoff et al. (2020), was adopted as the default model. For Fed-ECHO, we utilized a 2D U-net model, following the implementation from Ronneberger et al. (2015). The number of institutions involved in federated training for each task is listed in Appendix D. Our experiments mainly focus on the multi-center FL scenario (i.e., cross-silo), where all institutions participate in training at each communication round. Considering the trade-off between computation and communication, we set the local training epoch to 1 and the communication rounds to 50 throughout experiments except Fed-Consist. Since Fed-Consist requires extra rounds for training on clients with full labels before starting federated learning, we set the communication rounds to 100, where 50 rounds are for labeled clients training and another 50 rounds are normal FL training.

**Evaluation Strategies.** For a comprehensive evaluation, we build a local and global evaluation set for both datasets in FedCVD. For the local one, we divide each local data into train/test sets by 8:2. For the global one, we collect each local test set together. Our experiments test all algorithms using two evaluation strategies: 1) Global test performance (GLOBAL) is evaluated on the global test set and used to determine whether the model has learned knowledge from other clients in the FL setting. The better results of GLOBAL indicate that the model is closer to the centralized training. 2) Local test performance (LOCAL) is evaluated on each local test set. The LOCAL is more practical in real-world applications than GLOBAL because it indicates performance improvement for its task without centralizing all local data.

## 4.2 BENCHMARK ON FED-ECG

The proposed Fed-ECG dataset poses significant challenges for FL scenarios, namely *non-IID data* and *long-tailed distribution*. We first compared the overall performance of mainstream FL

Table 3: The performance of different FL methods on Fed-ECG is reported using two metrics: Micro F1-Score (Mi-F1) and Mean Average Precision (mAP), both expressed as percentages (%). The best results for each configuration are highlighted in **bold**, while the second-best results are underlined.

| Methods | LOCAL | | | | | | | | GLOBAL | |
|---|---|---|---|---|---|---|---|---|---|---|
| | SPH | | PTB-XL | | SXPH | | G12EC | | | |
| | Mi-F1↑ | mAP↑ | Mi-F1↑ | mAP↑ | Mi-F1↑ | mAP↑ | Mi-F1↑ | mAP↑ | Mi-F1↑ | mAP↑ |
| SPH | 85.8±1.9 | 58.1±2.6 | 52.7±3.4 | 37.8±2.2 | 61.5±1.2 | 19.8±1.2 | 49.8±4.2 | 26.7±3.0 | 64.3±2.1 | 32.3±2.0 |
| PTB-XL | 69.9±0.5 | 38.9±0.3 | 76.8±0.9 | 55.7±0.5 | 26.3±0.8 | 22.7±0.3 | 42.2±0.8 | 31.6±0.6 | 50.4±0.3 | 35.9±0.7 |
| SXPH | 22.7±0.2 | 29.8±0.7 | 17.0±0.4 | 27.2±0.3 | 88.1±0.2 | 37.7±0.4 | 56.9±0.4 | 29.4±0.6 | 51.5±0.2 | 32.7±0.2 |
| G12EC | 23.7±2.0 | 31.7±2.7 | 24.7±3.3 | 30.5±1.5 | 61.6±5.5 | 25.3±2.1 | 72.3±10.2 | 38.5±2.8 | 44.7±4.3 | 29.3±2.5 |
| FedAvg | 69.0±10.1 | 58.5±1.2 | 50.3±5.3 | 54.4±0.5 | 77.6±0.7 | 37.2±0.3 | 66.3±0.9 | 39.5±0.5 | 67.9±3.8 | 50.8±0.4 |
| FedProx | 74.0±7.5 | 60.3±2.9 | 55.6±2.7 | 56.4±0.6 | 73.2±1.0 | 36.0±0.8 | 70.2±2.3 | **43.8±1.8** | 68.8±2.6 | **52.3±0.9** |
| Scaffold | 77.5±2.6 | 58.0±1.2 | 56.9±1.7 | 55.9±0.7 | 73.3±1.0 | 36.2±0.6 | 70.7±2.9 | 42.7±1.1 | **70.1±0.8** | 52.1±0.7 |
| FedInit | 73.0±6.6 | 58.2±0.7 | 54.1±5.2 | 55.6±1.3 | 73.5±0.5 | 36.6±0.1 | 67.8±2.0 | 41.5±1.0 | 68.1±3.0 | 51.5±0.9 |
| Ditto | 82.8±4.4 | **63.1±4.2** | **74.8±1.4** | **58.3±0.6** | 86.5±1.5 | **38.1±0.6** | **73.4±6.7** | 42.2±4.0 | 68.1±2.9 | 48.7±1.4 |
| FedSM | 77.2±7.2 | 58.8±1.3 | 59.1±4.5 | 56.4±1.4 | 69.8±0.8 | 35.0±0.5 | 67.7±3.6 | 42.9±2.4 | 68.9±2.5 | 51.2±0.7 |
| FedALA | **84.4±4.0** | 62.0±7.0 | 71.7±5.7 | 57.1±2.2 | **88.2±0.1** | 37.4±0.2 | 66.7±5.9 | 41.2±2.3 | 67.8±1.9 | 50.8±1.3 |
| Central. | 84.9±0.5 | 54.8±0.5 | 71.4±5.0 | 55.2±2.9 | 84.1±1.6 | 36.5±1.1 | 72.2±3.7 | 41.5±1.3 | 80.0±2.1 | 63.2±2.8 |

algorithms on Fed-ECG, with the evaluated local and global performance shown in Table 3. The results indicate that FL has advantages over local training. However, existing FL algorithms still lag behind centralized training.

To better illustrate the impact of these two challenges on FL performance, we conducted experiments under supplementary settings by employing the additional evaluation metrics presented in 3.3. For the *non-IID challenge*, we compared the performance differences between natural partitioning and two simulated partitions (random and non-IID), with the simulated non-IID partition described in Appendix D. Figure 3 compares the performance (percentage relative to centralized training) between FL algorithms trained under the three partitioning settings. The results reveal that Fed-ECG's natural partitioning poses significantly greater challenges compared to the two simulated partitions.

For the *long-tail challenge*, we used the mAP metric in Table 3 to evaluate the overall performance of algorithms across different classes. In general, FL algorithms designed for heterogeneous scenarios demonstrate an advantage in addressing long-tail issues, with personalized algorithms like Ditto and FedALA showing better results in local tests. However, in global tests, the FedProx algorithm more effectively handles long-tail problems. Comparisons with centralized training reveal that FL scenarios tend to amplify the impact of long-tail distributions.

With regard to the two metrics specifically designed for gauging the long-trail challenge, the GLOBAL F1-STD results of different FL algorithms are visually presented in Figure 4, showing a pattern consistent with Table 3 and underscoring the challenges posed by long-tail distributions. Table 4 presents the Top-K metrics for various K values, highlighting the significant long-tail characteristics of Fed-ECG. The results show that mainstream FL algorithms struggle to address long-tail issues effectively, performing worse compared to centralized training.

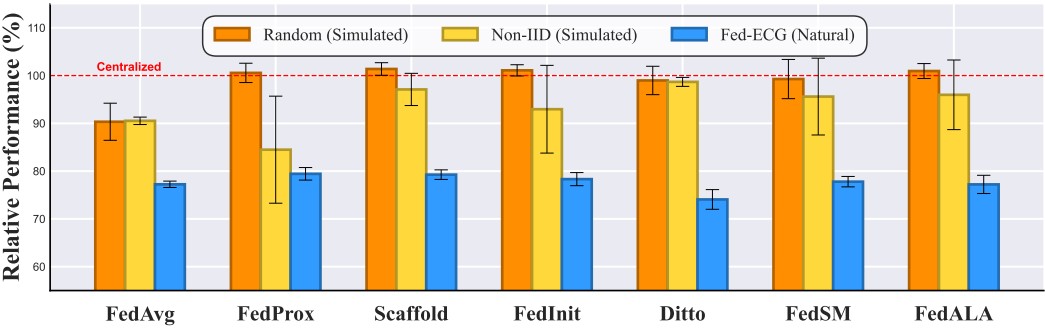

Figure 3: Demonstration of Fed-ECG's *non-IID challenge*: Comparisons of performance (relative Mean Average Score %) between artificial partitions (simulated random and non-IID partitions) and Fed-ECG's natural partition across different FL algorithms.

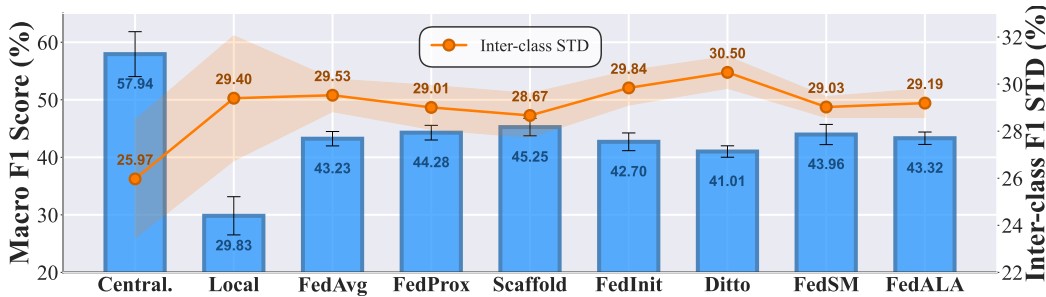

Figure 4: Demonstration of Fed-ECG's *long-tail challenge*: Average Macro F1-Score (%) and Standard Deviation across classes for various FL Algorithms.

Table 4: Demonstration of Fed-ECG's *long-tail challenge* through the performance (F1-Score (%)) differences (measured by relative performance drop) on head and tail class groups of varying sizes. "Top-K" denotes the selection of K classes with the most/fewest samples as the head/tail group. Comparisons are made among various FL algorithms, with the algorithm achieving the best result (minimum drop) highlighted in **bold** and the second-best results underlined.

| Method | Top-K F1 Performance | | | | | | | | | | | |
| | K=1(5%) | | | K=3(15%) | | | K=5(25%) | | | K=10(50%) | | |
| | Head | Tail | Drop | Head | Tail | Drop | Head | Tail | Drop | Head | Tail | Drop |
|---|---|---|---|---|---|---|---|---|---|---|---|---|
| FedAvg | 71.9±12.5 | 38.1±5.0 | 47.0 | 85.8±4.2 | 16.0±2.2 | 81.3 | **74.1±2.5** | 25.5±1.2 | 65.6 | 57.7±2.5 | 28.7±1.1 | 50.3 |
| FedProx | 76.8±4.5 | 30.8±2.6 | 59.9 | 87.6±1.4 | 13.1±1.2 | 85.0 | 71.2±0.9 | 22.7±2.6 | 68.1 | 57.5±1.8 | 31.1±2.2 | 46.0 |
| Scaffold | 77.4±3.2 | 33.8±4.9 | 56.4 | 87.7±1.2 | 14.6±2.2 | 83.4 | 71.3±0.7 | 24.1±1.4 | **66.3** | **58.4±1.7** | **32.1±2.3** | **45.0** |
| FedInit | 77.2±2.8 | 29.7±2.7 | 61.5 | 87.8±1.0 | 12.9±1.2 | 87.9 | 71.3±0.6 | 23.2±0.5 | 69.8 | 56.4±2.5 | 29.0±0.8 | 45.8 |
| Ditto | 73.5±7.4 | 23.8±5.8 | 67.6 | 86.4±2.5 | 10.2±1.7 | 88.2 | 70.6±2.1 | 21.4±1.1 | 69.7 | 54.4±2.6 | 27.6±1.7 | 49.3 |
| FedSM | 75.5±8.2 | 25.8±3.6 | 65.8 | 87.2±2.9 | 10.5±2.2 | 85.3 | 69.5±1.8 | 21.0±0.9 | 67.5 | 57.0±1.9 | 30.9±2.3 | 48.5 |
| FedALA | **72.4±5.4** | **38.9±5.8** | **46.2** | 86.0±1.8 | 16.2±2.1 | 81.1 | 73.7±1.5 | 25.2±1.2 | 65.7 | 57.9±1.7 | 28.8±1.0 | 50.3 |
| Central. | 88.6±2.3 | 35.6±5.9 | 59.8 | 92.4±1.6 | 19.5±7.0 | 78.9 | 84.1±1.9 | 29.9±5.5 | 64.5 | 71.3±3.6 | 44.5±4.4 | 37.6 |

## 4.3 BENCHMARK ON FED-ECHO

The proposed Fed-ECHO dataset presents one of the most challenging FL settings: *label incompleteness*, which can be viewed as an enhanced version of label-non-IID. Specifically, all annotated video frames from Institution 1 are completely segmented into four regions: BG, $LV_{Endo}$, $LV_{Epi}$, and LA. In contrast, Institution 2 and Institution 3 can only recognize the $LV_{Epi}$ and $LV_{Endo}$ regions in their annotated video frames, respectively, with the remaining regions simply labeled as "BG." For convenience, we refer to the BG labels from Institution 2 and Institution 3 as "Maybe-BG," indicating these segmentations may be unreliable. This discrepancy introduces potential conflicts between the "Maybe-BG" labels of Institution 2 and Institution 3 and the corresponding "reliable" labels of Institution 1, resulting in misleading labels that affect model convergence.

To mitigate the impact of misleading labels, we propose a straightforward baseline strategy, *supervised-only*. During supervised model training, we input the data from all three Institutions into the model without additional processing, allowing the model to benefit from the rich data features. However, when calculating the loss, we mask out the "Maybe-BG" regions in the video frames from Institution 2 and Institution 3. This means that for samples from Institution 2 and Institution 3, we only compute the training loss on the "reliable foreground". This strategy ensures the model learns segmentation capabilities from completely reliable labels. Additionally, during model segmentation performance evaluation, we also exclude the "Maybe-BG" regions from the test samples, preventing them from influencing the model's performance metrics.

Table 5 compares the performance of mainstream FL algorithms with centralized/isolated learning on Fed-ECHO, evaluated using Dice and Hausdorff distance ($d_H$). Except for the semi-supervised learning algorithms Centralized (semi-sup) and Fed-Consist, all algorithms use the previously mentioned supervised-only strategy. The results underscore the viability of FL in the Fed-ECHO setting, as most FL algorithms exhibit superior global performance compared to models trained independently by individual institutions. However, due to high degree of data heterogeneity, none of the evaluated FL algorithms outperform locally trained models on each client's test dataset, indicating a need for more personalized and heterogeneity-resistant FL strategies.

Table 5: The performance of different FL methods on Fed-ECHO, with DICE (%) and $d_H$ representing DICE index and Hausdorff distance respectively. The best results for each configuration are highlighted in **bold**, while the second-best results are underlined.

| Mthods | LOCAL | | | | | | GLOBAL | |
| --- | --- | --- | --- | --- | --- | --- | --- | --- |
| | CAMUS | | ECHONET-DYNAMIC | | HMC-QU | | | |
| | Dice↑ | $d_H$↓ | Dice↑ | $d_H$↓ | Dice↑ | $d_H$↓ | Dice↑ | $d_H$↓ |
| CAMUS | 88.2±0.8 | 5.196±0.360 | 46.5±3.9 | 24.246±0.442 | 63.4±4.2 | 22.000±12.914 | 66.1±2.8 | 17.147±4.187 |
| ECHONET-DYNAMIC | 24.4±6.0 | 71.917±2.832 | 88.9±5.5 | 5.577±1.413 | - | - | 37.8±3.8 | 59.165±1.411 |
| HMC-QU | 15.8±1.0 | 76.368±0.988 | - | - | 94.1±0.7 | 7.110±2.900 | 36.6±0.3 | 61.159±0.931 |
| FedAvg | 26.2±3.7 | 48.343±8.719 | 56.4±8.8 | 33.127±10.721 | 67.9±3.5 | 34.004±5.287 | 50.2±5.3 | 38.491±8.058 |
| FedProx | 74.8±18.7 | 13.928±11.742 | **82.3±3.5** | 13.402±2.975 | 66.7±12.4 | 16.181±16.329 | 74.6±11.4 | 14.504±9.134 |
| Scaffold | 81.5±2.1 | 9.981±2.482 | 81.0±2.1 | 12.543±2.157 | **74.6±2.1** | 7.551±0.885 | 79.0±0.7 | 10.025±1.467 |
| FedInit | 83.5±0.9 | 7.799±0.665 | 81.6±2.2 | **12.240±1.091** | 73.4±3.0 | **7.542±0.918** | **79.5±0.5** | **9.193±0.558** |
| Ditto | **88.2±0.4** | **4.796±0.085** | 56.9±3.3 | 28.381±4.043 | 56.3±2.2 | 27.321±15.627 | 78.1±1.8 | 10.658±2.372 |
| FedSM | 80.2±6.0 | 11.339±5.868 | 81.1±1.5 | 12.580±1.288 | 72.7±2.0 | 10.913±4.128 | 78.0±2.2 | 11.611±2.308 |
| FedALA | 80.5±1.6 | 8.700±1.245 | 51.3±2.4 | 36.472±2.686 | 47.1±0.9 | 52.128±4.356 | 52.3±2.0 | 36.811±2.630 |
| Fed-Consist | 85.9±0.2 | 11.904±0.442 | 75.2±0.9 | 27.480±1.440 | 66.3±0.2 | 34.037±1.777 | 75.8±0.3 | 24.474±1.155 |
| FedPSL | 53.5±9.3 | 37.277±9.166 | 77.0±2.9 | 12.873±1.589 | 67.8±14.1 | 29.166±15.660 | 66.1±7.5 | 26.439±7.831 |
| Central.(sup) | 89.9±0.4 | 4.643±0.097 | 48.5±22.2 | 43.684±19.659 | 65.0±14.6 | 30.557±14.831 | 67.8±12.1 | 26.295±11.379 |
| Central.(ssup) | 90.3±0.2 | 3.872±0.067 | 91.7±0.5 | 4.370±0.181 | 91.1±1.7 | 3.005±0.732 | 91.0±0.6 | 3.749±0.242 |

On the global test set, FL algorithms specifically designed to address heterogeneity, such as FedInit and Scaffold, consistently demonstrate significant advantages over simpler algorithms like FedAvg. Notably, these algorithms also outperform the Centralized (sup) model, which we attribute to FL effectively mitigating the impact of intra-batch heterogeneity(e.g., within the same batch, there are four-label data from Institution 1 and single-label data from Institution 2 or Institution 3).

Additionally, to leverage the substantial amount of partially labeled data from Institution 2 and Institution 3 and potentially mitigate label heterogeneity, we introduced semi-supervised learning algorithms for comparison. These include the centralized semi-supervised model, Centralized (ssup), and federated semi-supervised algorithms such as Fed-Consist and FedPSL. The Centralized (ssup) model significantly outperformed its fully supervised counterpart, underscoring the value of utilizing unlabeled video frames. Similarly, Fed-Consist outperformed FedAvg and FedProx, although it still exhibited a noticeable performance gap compared to the centralized semi-supervised algorithm and lagged behind fully supervised FL algorithms like Scaffold and FedInit. While FedPSL performed well on certain participating client, it showed greater instability overall, largely due to its sensitivity to client-side feature heterogeneity.

Therefore, the highly heterogeneous Fed-ECHO scenario poses significant challenges for FL algorithms, requiring them to adapt to heterogeneous data and effectively leverage unlabeled data across different data domains.

## 5 CONCLUSION

This paper has introduced FedCVD, the first real-world multi-center FL benchmark for CVD data, which consists of two datasets and their respective tasks: Fed-ECG and Fed-ECHO. It presents three major challenges due to the heterogeneous distribution of real-world data: non-IID, long-tailed labels, and label incompleteness. We conducted extensive comparative and validation experiments, testing mainstream FL algorithms and centralized training on these tasks. Experimental results show that the natural non-IID characteristics in FedCVD are more challenging than the manually partitioned setups in most previous federated benchmarks, and mainstream algorithms perform poorly in the long-tail tests of FedCVD. For the most difficult task, i.e., the label-incomplete Fed-ECHO, mainstream FL algorithms barely maintain utility but are still better than non-cooperative algorithms that only utilize unlabeled data on each client. Federated semi-supervised learning algorithms that leverage unlabeled data achieve some performance improvement. Beyond, as a flexible and extensible framework, FedCVD is meant to be a step towards developing FL in the CVD domain.

**Limitations and Future Work.** FedCVD presents a realistic and challenging scenario that tests FL algorithms' ability to mitigate data heterogeneity, handle long-tailed classes, and utilize unlabeled data. However, FedCVD currently offers only two tasks and a limited variety of data types. Additionally, the FL algorithms compared in experiments, particularly semi-supervised ones, are limited. In future work, we will expand the data range of FedCVD, aiming for it to inspire future FL research in real-world medical contexts, especially with CVD data.

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

## A    BROADER IMPACT

Considering that this research exclusively involves the repurposing of existing open-source databases, the associated risks are limited. However, it is important to acknowledge that all datasets utilized in this study may be influenced by biases inherent in the original data collection processes, such as those related to gender, age, or race. Unfortunately, identifying the sources of potential biases is challenging because the data have been appropriately pseudonymized. Moreover, records such as electrocardiograms and echocardiograms cannot be easily linked to specific demographic attributes such as age, ethnicity, or gender by non-medical experts. Nonetheless, our work discloses certain metadata of the datasets, including geographical origin, gender distribution, and age distribution. This exposure may aid in identifying underlying geographical biases, which are anticipated in real-world federated learning scenarios.

While prioritizing simplicity and utility, the current benchmark does not include privacy metrics. Nevertheless, privacy remains critically important in the cardiovascular disease domain, and we strongly encourage the research community to address these considerations. Thanks to the modularity of FedCVD, we can add privacy components easily. Therefore, we anticipate that FedCVD will address privacy concerns related to federated learning within the cardiovascular disease domain in the future.

## B    DATASETS REPOSITORY AND MAINTENANCE PLANE

### B.1    DATASET REPOSITORY.

The code is now available at `https://anonymous.4open.science/r/ZYNTMBB-8848`. Considering licenses, users need to download the data manually through the original dataset link.

### B.2    MAINTENANCE PLAN

We shall adhere to a maintenance plan to uphold the integrity of the codebase and ensure the conformity of supplied datasets to requisite standards. In particular, this maintenance plan encompasses:

- Fixing bugs affecting the correctness of our code, whether identified by the community or ourselves;
- Introducing additional variants of federated learning techniques, including alternative methods within the scope of cross-silo federated learning and federated semi-supervised learning methodologies;
- Adding new functional modules, such as privacy protection components.
- Regarding datasets, reviewing potential updates of the datasets referenced in the FedCVD, including but not limited to introducing new tasks or modalities;

## C    DESCRIPTION OF USING ALGORITHMS

**Semi-supervised**: Follows a pseudo-labeling approach by Lee et al. (2013). Specifically, the model was initially trained exclusively on the labeled data for 10 rounds. Subsequently, pseudo-labels were generated for the unlabeled data and incorporated into the training process for an additional 40 rounds. During this phase, we dynamically adjusted the weight of the loss function for the unlabeled data using a parameter $\alpha$, ensuring a gradual and adaptive integration of pseudo-labeled data into the training pipeline.

**FedAvg**: Implements a simple weighted average of model parameters from all participating clients during aggregation, without additional constraints.

**FedProx**: Introduces a regularization term that penalizes the divergence between the local and global models during training, mitigating the challenges posed by non-IID data distributions.

**Scaffold**: Utilizes control variates and server-side learning rate adjustments to reduce the impact of non-IID client updates, enhancing convergence and stability.

Table 6: Overview of the datasets, tasks, metrics and baseline models in FedCVD.

| Dataset | Fed-ECG | | | | Fed-ECHO | | |
|---|---|---|---|---|---|---|---|
| Task Type | Multi-label Classification | | | | 2D Segmentation | | |
| Input | 12-lead ECG Signal | | | | Echocardiogram | | |
| Prediction (y) | Diagnostic Statement | | | | Cardiac Structure Mask | | |
| Data source | SPH | PTB-XL | SXPH | G12EC | CAMUS | ECHONET-DYNAMIC | HMC-QU |
| Original Patient Size | 24,666 | 18,885 | 45,152 | UNKNOWN | 500 | 10,030 | 109 |
| Original Sample Size | 25,770 | 21,837 | 45,152 | 10,344 | 1000 | 20,060 | 2,349 |
| Preprocessing | Label Alignment | | | | Resizing and Label Alignment | | |
| Patient Size | 21,530 | 16,699 | 36,272 | UNKNOWN | 500 | 10,024 | 109 |
| Sample Size | 22,425 | 19,019 | 36,272 | 6,205 | 1000 | 20,048 | 2,349 |
| Model | ResNet | | | | U-net | | |
| Metrics | Micro F1 / mAP | | | | DICE / Hausdorff distance | | |
| Input Dimension | $12 \times 5000$ | | | | $112 \times 112$ | | |

**FedInit**: Employs a relaxed, personalized initialization at the beginning of each local training phase, addressing the disparity caused by non-IID data across clients.

**Ditto**: Implements personalized federated learning by adding a regularization term to minimize the gap between the local personalized model and the global model, ensuring both global consistency and local adaptability.

**FedSM**: Combines local models, a global model, and a model selector to achieve personalized federated learning. The selector improves the accuracy and adaptability of the federated learning process.

**FedALA**: Adaptively aggregates global and local models at selected layers to address client heterogeneity, improving training efficiency.

**Fed-Consist**: Initially trains a model with data from institutions with complete labels. For institutions with incomplete labels, pseudo-labels are generated using the global model on both raw and augmented data. Only instances with both predictions exceeding a confidence threshold are retained as final pseudo-labels, utilizing a consistency-based semi-supervised approach.

**FedPSL**: Splits the model into a task-agnostic feature extractor and a task-dependent classifier. The feature extractor is aggregated across all clients using a FedAvg-like approach, while the classifier is aggregated only among clients sharing the corresponding label.

# D    FED-ECG

## D.1    DESCRIPTION

Fed-ECG consists of four datasets: SPH, PTB-XL, SXPH, and G12EC. The order of leads of each dataset is I, II, III, aVR, aVL, aVF, V1, V2, V3, V4, V5, V6. The overview of Fed-ECG is shown in Table 6. Table 7 shows demographics information for four datasets in Fed-ECG.

**SPH.**    The original Shandong Provincial Hospital (SPH) database contains 25,770 12-lead ECG records from 24,666 patients, which were acquired from Shandong Provincial Hospital between 2019/08 and 2020/08. The record length is between 10 and 60 seconds. The sampling frequency is 500 Hz. All ECG records are in full compliance with the AHA standard, which aims for the standardization and interpretation of the electrocardiogram and consists of 44 primary statements and 15 modifiers as per the standard. 46.04% records in this dataset contain ECG abnormalities. Moreover, 14.45% records have multiple diagnostic statements.

**PTB-XL.**    The original PTB-XL database contains 21,837 12-lead ECG records from 18,885 patients of 10 seconds length at the Physikalisch Technische Bundesanstalt (PTB) between October 1989 and June 1996. The original records are resampled to both 100 Hz and 500 Hz. For consistency, we only use the records whose frequency is 500 Hz. Each data is annotated by up to two cardiologists with the SCP-ECG standard.

**SXPH.** This database contains 12-lead ECGs of 45,152 patients with a 500 Hz sampling rate under the auspices of Chapman University, Shaoxing People's Hospital (Shaoxing Hospital Zhejiang University School of Medicine), and Ningbo First Hospital. The record length is 10 seconds. All records are labeled by professional experts with the SNOMED-CT standard.

**G12EC.** This Georgia 12-lead ECG Challenge (G12EC) database is provided by the Physi-oNet/Computing in Cardiology Challenge 2020. Only 10,344 training data from this database are open to the public. The record length is not longer than 10 seconds with a sample frequency of 500 Hz. All records are labeled with the SNOMED-CT standard as well.

Table 7: Demographics information for Fed-ECG.

| Client | Sex | Dataset size | Age | Age Range |
|--------|------|------------|-----------------|---------|
| Client1 | Female | 9,502 | $48.73 \pm 15.67$ | 18 - 92 |
|  | Male | 12,923 | $50.35 \pm 15.49$ | 18 - 95 |
| Client2 | Female | 8,930 | $59.80 \pm 18.42$ | 3 - 89 |
|  | Male | 10,089 | $58.40 \pm 15.66$ | 2 - 89 |
| Client3 | Female | 14,830 | $58.36 \pm 20.11$ | 4 - 89 |
|  | Male | 21,442 | $60.28 \pm 19.10$ | 4 - 89 |
| Client4 | Female | 2,668 | $61.37 \pm 16.51$ | 20 - 89 |
|  | Male | 3,537 | $61.35 \pm 15.04$ | 14 - 89 |

### D.2 LICENSE AND ETHICS

All four databases are open-access. The SPH database is open access at Figshare, while the rest databases are open access at PhysioNet under a Creative Commons Attribution 4.0 International Public License.

The PTB-XL database was supported by the Bundesministerium für Bildung und Forschung (BMBF) through the Berlin Big Data Center under Grant 01IS14013A and the Berlin Center for Machine Learning under Grant 01IS18037I and by the EMPIR project 18HLT07 MedalCare. The EMPIR initiative is cofunded by the European Union's Horizon 2020 research and innovation program and the EMPIR Participating States.

The institutional review board of Shaoxing People's Hospital and Ningbo First Hospital of Zhejiang University approved the study of the SXPH database, granted the waiver application to obtain informed consent, and allowed the data to be shared publicly after de-identification. The requirement for patient consent was waived.

### D.3 DOWNLOAD AND PREPROCESSING

### D.3.1 DOWNLOAD

The four datasets can be downloaded using the URLs below:

1. **SPH:** `https://springernature.figshare.com/collections/A_large-scale_multi-label_12-lead_electrocardiogram_database_with_standardized_diagnostic_statements/5779802/1`

2. **PTB-XL:** `https://physionet.org/content/ptb-xl/1.0.3/`

3. **SXPH:** `https://physionet.org/content/ecg-arrhythmia/1.0.0/`

4. **G12EC:** `https://physionet.org/content/challenge-2020/1.0.2/`

### D.3.2 PREPROCESSING

Raw 12-lead ECG signals have varying sequence lengths and raw 12-lead ECG signals have varying sequence lengths and annotated standards which must be standardized before FL training. Therefore, we first set a signal length to 10 seconds. We pad the signal with edge value at the edge for those whose length is shorter than 10 seconds and cut off the signal at 10 seconds for those whose length is

Table 8: Label relationship between original label and ours.

| ours | Original Label | | | |
|---|---|---|---|---|
| | SPH | PTB-XL | SXPH | G12EC |
| NORM (Normal) | Normal | Normal | - | - |
| STACH (Sinus tachycardia) | Sinus tachycardia | Sinus tachycardia | Sinus tachycardia | 427084000 |
| SBRAD (Sinus bradycardia) | Sinus bradycardia | Sinus bradycardia | Sinus bradycardia | 426177001 |
| SARRH (Sinus arrhythmia) | Sinus arrhythmia | Sinus arrhythmia | - | 427393009 |
| PAC (Atrial premature complex(es)) | Atrial premature complex(es) | Atrial premature complex | - | - |
| AFIB (Atrial fibrillation) | Atrial fibrillation | Atrial fibrillation | Atrial fibrillation | 164889003 |
| AFLT (Atrial flutter) | Atrial flutter | Atrial flutter | Atrial flutter | 164890007 |
| SVTAC (Supraventricular tachycardia) | - | Supraventricular tachycardia | Supraventricular tachycardia | 426761007 |
| PVC (Ventricular premature complex) | Ventricular premature complex(es) | Ventricular premature complex | - | 164884008 |
| 1AVB (First degree AV block) | - | First degree AV block | 1 degree atrioventricular block | 270492004 |
| 2AVB (Second degree AV block) | Second-degree AV block, Mobitz type I (Wenckebach) | | 2 degree atrioventricular block(Type one) | 54016002 |
| | Second-degree AV block, Mobitz type II | | 2 degree atrioventricular block(Type two) | 28189009 |
| | 2:1 AV block | Second degree AV block | | 164903001 |
| | AV block, varying conduction | | 2 degree atrioventricular block | 195042002 |
| | AV block, advanced (high-grade) | | | 284941000119107 |
| 3AVB (Third degree AV block) | AV block, complete (third-degree) | Third degree AV block | 3 degree atrioventricular block | 27885002 |
| LBBB (Left bundle branch block) | Left anterior fascicular block | Left anterior fascicular block | | 445118002 |
| | Left posterior fascicular block | Left posterior fascicular block | Left bundle branch block | 445211001 |
| | Left bundle-branch block | Complete left bundle branch block | | 164909002 |
| RBBB (Right bundle branch block) | Incomplete right bundle-branch block | Incomplete right bundle branch block | | 713426002 |
| | | | Right bundle branch block | 59118001 |
| | Right bundle-branch block | Complete right bundle branch block | | 164907000 |
| LAO/LAE (Left atrial overload/enlargement) | Left atrial enlargement | Left atrial overload/enlargement | - | 67741000119109 |
| LVH (Left ventricular hypertrophy) | Left ventricular hypertrophy | Left ventricular hypertrophy | - | 164873001 |
| RVH (Right ventricular hypertrophy) | Right ventricular hypertrophy | Right ventricular hypertrophy | - | - |
| AMI (Anterior myocardial infarction) | Anterior MI | Anterior myocardial infarction | - | - |
| IMI (Inferior myocardial infarction) | Inferior MI | Inferior myocardial infarction | - | - |
| ASMI (Anteroseptal myocardial infarction) | Anteroseptal MI | Anteroseptal myocardial infarction | - | - |

longer than 10 seconds. Next, we only save the records whose label occurs in at least two databases. Finally, we align the labels of records in different databases. The relationship between the original label and our label is shown in Table8.

## D.4 BASELINE, LOSS FUNCTION AND EVALUATION

**Baseline Model.**    We implement a ResNet1d model with 34 layers. The final layer output is passed through a sigmoid function to encode the probability that each label corresponds to one 12-lead ECG signal.

**Loss function.**    The model was directly trained for the Binary CrossEntropy Loss (BCELoss), defined as:

$$\text{BCE}(\mathbf{y}, \hat{\mathbf{y}}) = -[\sum_{i=1}^{n} y_i \log(\hat{y}_i) + \sum_{i=1}^{n} (1 - y_i) \log(1 - \hat{y}_i)] \tag{1}$$

**Evaluation Metrics.**    In multi-label classification for Fed-ECG, the micro F1 score is used as the main metric to evaluate the performance of the model. Given $N$ labels, the micro-precision ($P_{\text{micro}}$) and micro-recall ($R_{\text{micro}}$) are calculated as $P_{\text{micro}} = \frac{\sum_{i=1}^{N} \text{TP}_i}{\sum_{i=1}^{N} (\text{TP}_i + \text{FP}_i)}$ and $R_{\text{micro}} = \frac{\sum_{i=1}^{N} \text{TP}_i}{\sum_{i=1}^{N} (\text{TP}_i + \text{FN}_i)}$, where $\text{TP}_i$ is the number of true positives for label $i$, $\text{FP}_i$ is the number of false positives for label $i$, $\text{FN}_i$ is the number of false negatives for label $i$. The micro F1 score ($F1_{\text{micro}}$) is then calculated as:

$$F1_{\text{micro}} = \frac{2 \cdot P_{\text{micro}} \cdot R_{\text{micro}}}{P_{\text{micro}} + R_{\text{micro}}} \tag{2}$$

For Fed-ECG's Multi-Label Classification task, the Mean Average Precision (mAP) is adopted to measure the classification performance across all labels (including long-tailed labels), calculated by averaging the average precision (AP) for each label, defined as:

$$\text{mAP} = \frac{1}{L} \sum_{i=1}^{L} \sum_{k=1}^{n} P_i(k) \Delta r_i(k) \tag{3}$$

where $L$ is the total number of labels, and $\text{AP}_i$ is the average precision for the $i$-th label, $P_i(k)$ is the precision for label $i$ at the $k$-th threshold, and $\Delta r_i(k)$ is the change in its recall at the $k$-th threshold.

## D.5 TRAINING DETAIL

**Optimization parameters.**    We optimize the ResNet1d using SGD optimizer, with a batch size of 32. We train our model for 50 epochs on one NVIDIA A100-PCIE-40GB. To ensure robustness and statistical reliability, we repeat each experiment five times and report the mean and standard deviation of the results.

**Hyperparameter Search**  For centralized and local model training, we first conduct a search for optimal learning rates from the set {1e-5, 1e-4, 1e-3, 1e-2, 1e-1} during centralized model training. The learning rate that yields the best micro-F1 score is then used for local model training. For the federated learning strategies, we employ the following hyperparameter grid:

- For clients' learning rates (all strategies): {1e-5, 1e-4, 1e-3, 1e-2, 1e-1}.

- For server size learning rate (Scaffold strategy only): {1e-2, 1e-1, 1.0}.

- For FedProx and Ditto strategies, the parameter $\mu$ is selected from {1e-2, 1e-1, 1.0}.

- For FedInit, the parameter $\beta$ is chosen from {1e-1, 1e-2, 1e-3}.

- For FedSM, the parameters $\gamma$ and $\lambda$ are set to values from {0, 0.1, 0.7, 0.9} and {0.1, 0.3, 0.5, 0.7, 0.9}, respectively.

- For FedALA, the parameters layer index, $\eta$, threshold, and num_per_loss are fixed at 1, 1.0, 0.1, and 10, respectively, while rand_percent is selected from {5, 50, 80}.

Table 9: Hyperparameters used for the Fed-ECG with ResNet model.

| Methods | learning rate | optimizer | learning rate server | mu | beta | lambda | gamma | rand_percent |
|---|---|---|---|---|---|---|---|---|
| Central. | 0.1 | torch.optim.SGD | - | - | - | - | - | - |
| FedAvg | 0.1 | torch.optim.SGD | - | - | - | - | - | - |
| FedProx | 0.1 | torch.optim.SGD | - | 0.01 | - | - | - | - |
| Scaffold | 0.1 | torch.optim.SGD | 1.0 | - | - | - | - | - |
| FedInit | 0.1 | torch.optim.SGD | 1.0 | - | 0.01 | - | - | - |
| Ditto | 0.1 | torch.optim.SGD | - | 0.01 | - | - | - | - |
| FedSM | 0.1 | torch.optim.SGD | 1.0 | - | - | 0.1 | 0 | - |
| FedALA | 0.1 | torch.optim.SGD | 1.0 | - | - | - | - | 80 |

Table 10: Hyperparameters used for the Fed-ECG with Transformer model.

| Methods | learning rate | optimizer | learning rate server | mu | beta | lambda | gamma | rand_percent |
|---|---|---|---|---|---|---|---|---|
| Central. | 0.0001 | torch.optim.Adam | - | - | - | - | - | - |
| FedAvg | 0.0001 | torch.optim.Adam | - | - | - | - | - | - |
| FedProx | - | torch.optim.Adam | - | - | - | - | - | - |
| Scaffold | 0.0001 | torch.optim.Adam | 1.0 | - | - | - | - | - |
| FedInit | 0.0001 | torch.optim.Adam | 1.0 | - | 0.1 | - | - | - |
| Ditto | - | torch.optim.Adam | - | - | - | - | - | - |
| FedSM | 0.0001 | torch.optim.Adam | 1.0 | - | - | 0.3 | 0 | - |
| FedALA | 0.0001 | torch.optim.Adam | 1.0 | - | - | - | - | 5.0 |

**Non-IID partition.**  For the non-IID partition, we first pool the training data from the four clients. Then, we cluster the samples into 10 categories based on the cosine similarity and order them according to the number of samples contained in each category. Next, the sorted samples are divided into 32 shards. finally, 8 random shards are distributed to one client. The label distribution of each client with the non-IID partition is shown in Figure 5.

### D.6  SUPPLEMENTARY EXPERIMENT RESULTS

We provide additional evaluation metrics here. Table 11 presents an extensive array of evaluation metrics for various federated learning approaches applied to Fed-ECG. The Micro F1-Score (Mi-F1) and Hamming Loss (HL) serve as indicators of the overall performance, given their insensitivity to long-tail distributions. In contrast, the mean Average Precision score (mAP) provides insight into the average performance across individual labels. In addition, Table 12 presents the F1 score for each label, which more clearly demonstrates the impact of the long-tail distribution on each label.

Result (click "Generate" to refresh) Copy to clipboard

We conduct pairwise t-tests on the performance metrics **mAP** and **Micro-F1**, with the resulting p-values presented in Figure 13. The findings reveal the following:

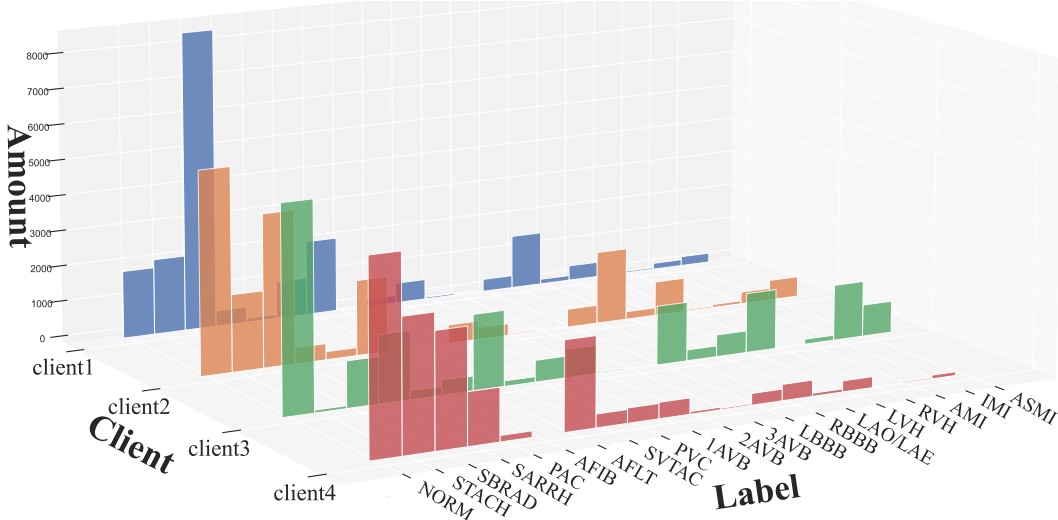

Figure 5: Label non-IID of the Fed-ECG dataset with the artificially non-IID partition, shown as the variation in the number of each label (right axis) across different clients (left axis).

Table 11: The performance of different FL methods on Fed-ECG, with Mi-F1, mAP, and HL representing Micro F1-Score, mean Average Precision score, and Hamming Loss, respectively. All metrics are present in percentage (%). The best results for each configuration are highlighted in **bold**, while the second-best results are underlined.

| Methods | SPH Mi-F1↑ | SPH mAP↑ | SPH HL↓ | PTB-XL Mi-F1↑ | PTB-XL mAP↑ | PTB-XL HL↓ | SXPH Mi-F1↑ | SXPH mAP↑ | SXPH HL↓ | G12EC Mi-F1↑ | G12EC mAP↑ | G12EC HL↓ | GLOBAL Mi-F1↑ | GLOBAL mAP↑ | GLOBAL HL↓ |
|---|---|---|---|---|---|---|---|---|---|---|---|---|---|---|---|
| SPH | 85.8 ±1.9 | 58.1 ±2.6 | 1.5 ± 0.2 | 52.7 ±3.4 | 37.8 ±2.2 | 5.8 ± 0.4 | 61.5 ±1.2 | 19.8 ±1.2 | 4.4 ± 0.1 | 49.8 ±4.2 | 26.7 ±3.0 | 6.4 ± 0.6 | 64.3 ±2.1 | 32.3 ±2.0 | 4.1 ± 0.2 |
| PTB-XL | 69.9 ±50.0 | 38.9 ±30.0 | 3.2 ± 0.1 | 76.8 ±90.0 | 55.7 ±50.0 | 3.1 ± 0.1 | 26.3 ±80.0 | 22.7 ±30.0 | 9.0 ± 0.2 | 42.2 ±80.0 | 31.6 ±60.0 | 8.1 ± 0.1 | 50.4 ±30.0 | 35.9 ±70.0 | 6.1 ± 0.1 |
| SXPH | 22.7 ±0.2 | 29.8 ±0.7 | 8.2 ± 0.0 | 17.0 ±0.4 | 27.2 ±0.3 | 10.3 ± 0.1 | 88.1 ±0.2 | 37.7 ±0.4 | 1.3 ± 0.0 | 56.9 ±0.4 | 29.4 ±0.6 | 5.4 ± 0.1 | 51.5 ±0.2 | 32.7 ±0.2 | 5.5 ± 0.0 |
| G12EC | 23.7 ±2.0 | 31.7 ±2.7 | 8.4 ± 0.9 | 24.7 ±3.3 | 30.5 ±1.5 | 10.1 ± 1.2 | 61.6 ±5.5 | 25.3 ±2.1 | 5.0 ± 1.2 | 72.3 ±10.2 | 38.5 ±2.8 | 4.1 ± 1.8 | 44.7 ±4.3 | 29.3 ±2.5 | 7.0 ± 1.1 |
| FedAvg | 69.0 ±10.1 | 58.5 ±1.2 | 3.4 ± 1.1 | 50.3 ±5.3 | 54.4 ±0.7 | 6.2 ± 0.9 | 77.6 ±0.7 | 37.2 ±0.3 | 2.5 ± 0.1 | 66.3 ±0.9 | 39.5 ±0.5 | 4.2 ± 0.1 | 67.9 ±3.8 | 50.8 ±0.4 | 3.7 ± 0.5 |
| FedProx | 74.0 ±7.5 | 60.3 ±2.9 | 2.9 ± 1.0 | 55.6 ±2.7 | 56.4 ±0.6 | 5.5 ± 0.5 | 73.2 ±1.0 | 36.0 ±0.8 | 3.0 ± 0.1 | 70.2 ±2.3 | **43.8** ±**1.8** | 3.8 ± 0.3 | 68.8 ±2.6 | **52.3** ±**0.9** | 3.6 ± 0.4 |
| Scaffold | 77.5 ±2.6 | 58.0 ±1.2 | 2.3 ± 0.2 | 56.9 ±1.7 | 55.9 ±0.7 | 5.2 ± 0.2 | 73.3 ±1.0 | 36.2 ±0.6 | 3.0 ± 0.1 | 70.7 ±2.9 | 42.7 ±1.1 | 3.7 ± 0.3 | **70.1** ±**0.8** | 52.1 ±0.7 | **3.4** ± **0.1** |
| FedInit | 73.0 ± 6.6 | 58.2 ± 0.7 | 3.1 ± 1.0 | 54.1 ± 1.8 | 55.6 ± 1.3 | 5.9 ± 0.9 | 73.5 ± 0.5 | 36.6 ± 0.1 | 3.0 ± 0.1 | 67.8 ± 2.0 | 41.5 ± 1.0 | 4.1 ± 0.3 | 68.1 ±3.0 | 51.5 ±0.9 | 3.8 ±0.5 |
| Ditto | 82.8 ±4.4 | **63.1** ±**4.2** | 1.8 ± 0.4 | **74.8** ±**1.4** | **58.3** ±**0.6** | **3.5** ± 0.2 | 86.5 ±1.5 | **38.1** ±**0.6** | 1.5 ± 0.2 | 73.4 ±6.7 | 42.2 ±4.0 | **3.6** ± **0.9** | 68.1 ±2.9 | 48.7 ±1.4 | 3.6 ± 0.3 |
| FedSM | 77.2 ± 7.2 | 58.8 ± 1.3 | 2.3 ± 0.6 | 59.1 ± 4.5 | 56.4 ± 1.4 | 5.1 ± 0.5 | 69.8 ± 0.8 | 35.0 ± 0.5 | 3.5 ± 0.1 | 67.7 ± 3.6 | 42.9 ± 2.4 | 4.1 ± 0.4 | 68.9 ± 2.5 | 51.2 ± 0.7 | 3.6 ± 0.3 |
| FedALA | **84.4** ± **4.0** | 62.0 ± 7.0 | **1.6** ± **0.4** | 71.7 ± 5.7 | 57.1 ± 2.2 | 3.8 ± 0.6 | **88.2** ± **0.1** | 37.4 ± 0.2 | **1.3** ± **0.0** | 66.7 ± 5.9 | 41.2 ± 2.3 | 4.4 ± 0.7 | 67.8 ± 1.9 | 50.8 ± 1.3 | 3.7 ± 0.3 |
| Central. | 84.9 ±0.5 | 54.8 ±0.5 | 1.6 ± 0.1 | 71.4 ±5.0 | 55.2 ±2.9 | 3.8 ± 0.6 | 84.1 ±1.6 | 36.5 ±1.1 | 1.7 ± 0.2 | 72.2 ±3.7 | 41.5 ±1.3 | 3.6 ± 0.3 | 80.0 ±2.1 | 63.2 ±2.8 | 2.3 ± 0.2 |

Table 12: The performance of different FL algorithms (F1 %) on each label of Fed-ECG. The best results for each label are marked in **bold**.

| label | client1 | client2 | client3 | client4 | fedavg | fedprox | scaffold | ditto | fedinit | fedsm | fedala | pooled |
|---|---|---|---|---|---|---|---|---|---|---|---|---|
| NORM | 79.8 ± 0.6 | 62.6 ± 0.5 | 0 | 0 | 71.9 ± 12.5 | 76.8 ± 4.5 | **77.4 ± 3.2** | 77.2 ± 2.8 | 73.5 ± 7.4 | 75.5 ± 8.2 | 72.4 ± 5.4 | 88.6 ± 2.3 |
| SBRAD | 88.0 ± 0.8 | 20.3 ± 2.0 | 88.2 ± 0.4 | 81.2 ± 11.5 | 90.4 ± 0.3 | 90.6 ± 0.6 | 90.4 ± 0.7 | **90.9 ± 0.1** | 90.8 ± 0.3 | 90.6 ± 0.3 | 90.7 ± 0.1 | 92.8 ± 2.9 |
| STACH | 87.9 ± 2.2 | 87.9 ± 1.4 | 90.5 ± 0.3 | 85.3 ± 4.8 | 95.2 ± 0.4 | 95.4 ± 0.5 | 95.3 ± 0.6 | 95.2 ± 0.3 | 94.9 ± 0.6 | **95.5 ± 0.5** | 94.8 ± 0.4 | 95.7 ± 0.8 |
| AFLT | 13.9 ± 0.9 | 9.7 ± 5.4 | 73.8 ± 0.4 | 18.2 ± 5.1 | **47.7 ± 3.0** | 26.1 ± 3.5 | 26.6 ± 2.6 | 27.0 ± 3.0 | 27.3 ± 2.7 | 19.6 ± 1.5 | 45.2 ± 4.0 | 73.2 ± 4.2 |
| RBBB | 60.6 ± 1.1 | 63.7 ± 0.9 | 40.3 ± 3.8 | 58.1 ± 9.5 | 65.0 ± 1.4 | **67.0 ± 0.8** | 66.8 ± 1.3 | 66.5 ± 0.7 | 66.5 ± 1.2 | 66.4 ± 1.4 | 65.3 ± 1.7 | 70.1 ± 1.5 |
| SARRH | 35.5 ± 4.2 | 2.6 ± 2.2 | 21.7 ± 0.2 | 17.2 ± 2.1 | 36.7 ± 5.8 | 40.0 ± 7.1 | **46.3 ± 1.7** | 39.6 ± 7.2 | 42.6 ± 6.1 | 46.0 ± 4.4 | 36.0 ± 4.5 | 59.8 ± 3.5 |
| AFIB | 44.7 ± 2.1 | 47.7 ± 0.6 | 13.7 ± 1.3 | 46.9 ± 0.9 | 52.2 ± 0.3 | 51.5 ± 0.9 | 52.2 ± 0.7 | 51.1 ± 1.1 | 51.1 ± 0.9 | 50.2 ± 0.9 | **52.4 ± 0.4** | 58.2 ± 1.1 |
| LVH | 10.4 ± 10.0 | 43.6 ± 0.6 | 21.6 ± 2.0 | 26.1 ± 5.8 | 26.3 ± 8.1 | **36.5 ± 10.0** | 36.3 ± 10.8 | 28.6 ± 12.6 | 10.9 ± 7.0 | 27.6 ± 15.4 | 29.0 ± 11.3 | 45.7 ± 17.3 |
| LBBB | 47.5 ± 14.3 | 57.0 ± 1.9 | 52.1 ± 1.8 | 47.8 ± 4.8 | 67.4 ± 1.1 | 66.9 ± 1.1 | 67.5 ± 1.0 | 66.8 ± 2.7 | 67.5 ± 1.1 | 66.6 ± 1.3 | **67.6 ± 0.7** | 68.7 ± 4.3 |
| IMI | 26.2 ± 3.5 | 44.8 ± 1.3 | 0 | 0 | 24.5 ± 3.4 | 24.3 ± 4.6 | 25.1 ± 6.3 | 20.9 ± 7.3 | 19.1 ± 4.8 | **32.5 ± 3.5** | 25.2 ± 2.7 | 60.6 ± 5.3 |
| 1AVB | 0 | 50.5 ± 4.1 | 67.2 ± 1.2 | 56.5 ± 10.1 | 64.6 ± 2.1 | 66.3 ± 1.0 | **66.7 ± 1.7** | 65.3 ± 0.9 | 60.8 ± 4.9 | 60.6 ± 4.8 | 64.0 ± 2.1 | 66.6 ± 3.1 |
| ASMI | 19.0 ± 5.6 | 41.4 ± 2.9 | 0 | 0 | 17.1 ± 5.7 | 24.4 ± 6.7 | 25.6 ± 4.4 | 21.4 ± 8.0 | 29.9 ± 10.3 | **35.4 ± 15.9** | 25.6 ± 7.0 | 68.2 ± 3.1 |
| PVC | 67.8 ± 2.2 | 62.4 ± 1.3 | 0 | 1.0 ± 1.0 | 53.4 ± 2.2 | 64.3 ± 1.4 | 63.6 ± 3.4 | 57.3 ± 4.8 | 56.7 ± 7.7 | **64.5 ± 6.0** | 51.7 ± 3.9 | 78.6 ± 3.5 |
| LAO/LAE | 0.6 ± 0.6 | 9.0 ± 2.3 | 0 | 36.2 ± 6.6 | 2.3 ± 1.6 | 17.3 ± 12.5 | 18.5 ± 10.3 | 5.4 ± 3.1 | 3.0 ± 4.2 | **18.5 ± 3.7** | 1.4 ± 1.3 | 34.6 ± 17.2 |
| PAC | 35.6 ± 2.2 | 4.4 ± 1.5 | 0 | 0 | 22.5 ± 5.6 | 25.0 ± 3.3 | **26.5 ± 4.5** | 25.0 ± 5.3 | 18.5 ± 3.2 | 25.0 ± 5.7 | 18.8 ± 5.4 | 48.0 ± 2.7 |
| SVTAC | 0 | 13.3 ± 6.8 | 78.4 ± 1.7 | 30.0 ± 9.2 | **79.1 ± 1.8** | 74.2 ± 9.5 | 76.5 ± 0.9 | 77.1 ± 1.2 | 75.9 ± 1.2 | 73.3 ± 4.3 | 77.6 ± 2.8 | 78.7 ± 0.8 |
| AMI | 5.3 ± 1.7 | 9.6 ± 2.6 | 0 | 0 | 0 | 0 | 0 | 0 | 0.4 ± 0.9 | 0 | 0 | 12.1 ± 9.5 |
| 2AVB | 5.2 ± 1.4 | 0 | 6.1 ± 3.7 | 0 | **10.0 ± 1.9** | 8.6 ± 2.1 | 10.0 ± 4.3 | 9.0 ± 2.4 | 6.8 ± 2.6 | 5.8 ± 3.7 | 9.7 ± 1.6 | 12.7 ± 5.8 |
| RVH | 0 | 14.6 ± 1.6 | 0 | 0 | 0 | 0 | 0 | 0 | 0 | 0 | 0 | 10.2 ± 13.4 |
| 3AVB | 13.3 ± 8.4 | 1.7 ± 3.5 | 40.2 ± 6.9 | 0 | 38.1 ± 5.0 | 30.8 ± 2.6 | 33.8 ± 4.9 | 29.7 ± 2.7 | 23.8 ± 5.8 | 25.8 ± 3.6 | **38.9 ± 5.8** | 35.6 ± 5.9 |

Table 13: The p-values of the pairwise t-tests on the performance, mAP (left) and Micro-F1 (right), of different FL algorithms on Fed-ECG, where those with significant performance differences detected are shown in **bold**.

| | FedProx | Scaffold | FedInit | Ditto | FedSM | FedALA |
|---|---|---|---|---|---|---|
| FedAvg | **0.009** | **0.016** | 0.201 | **0.048** | 0.462 | 0.983 |
| FedProx | - | 0.704 | **0.373** | **0.011** | **0.063** | **0.056** |
| Scaffold | - | - | 0.452 | **0.005** | 0.069 | **0.017** |
| FedInit | - | - | - | 0.058 | 0.646 | 0.531 |
| Ditto | - | - | - | - | **0.035** | **0.026** |
| FedSM | - | - | - | - | - | 0.535 |

| | FedProx | Scaffold | FedInit | Ditto | FedSM | FedALA |
|---|---|---|---|---|---|---|
| FedAvg | 0.73 | 0.27 | 0.94 | 0.93 | 0.71 | 0.99 |
| FedProx | - | 0.26 | 0.74 | 0.74 | 0.97 | 0.59 |
| Scaffold | - | - | 0.23 | 0.29 | 0.42 | 0.12 |
| FedInit | - | - | - | 0.98 | 0.63 | 0.91 |
| Ditto | - | - | - | - | 0.53 | 0.88 |
| FedSM | - | - | - | - | - | 0.63 |

Table 14: The performances of different FL methods on Fed-ECG with Transformer model, with Mi-F1 and mAP representing Micro F1-Score and mean average precision score, respectively. Both metrics are present in percentage (%).The best results for each configuration are highlighted in **bold**, while the second-best results are underlined.

| Methods | LOCAL | | | | | | | | GLOBAL | |
|---|---|---|---|---|---|---|---|---|---|---|
| | SPH | | PTB-XL | | SXPH | | G12EC | | | |
| | Mi-F1↑ | mAP↑ | Mi-F1↑ | mAP↑ | Mi-F1↑ | mAP↑ | Mi-F1↑ | mAP↑ | Mi-F1↑ | mAP↑ |
| SPH | 86.4 ± 0.4 | 52.8 ± 3.3 | 53.6 ± 0.7 | 36.6 ± 0.6 | 61.0 ± 1.4 | 20.7 ± 0.1 | 51.9 ± 1.0 | 27.5 ± 1.8 | 64.7 ± 0.7 | 32.9 ± 0.9 |
| PTB-XL | 69.3 ± 0.4 | 35.0 ± 0.9 | 73.7 ± 0.7 | 48.4 ± 1.4 | 26.6 ± 2.5 | 18.5 ± 1.5 | 40.4 ± 1.2 | 26.5 ± 0.5 | 49.3 ± 1.2 | 31.2 ± 1.6 |
| SXPH | 24.4 ± 1.0 | 26.7 ± 1.6 | 19.2 ± 1.2 | 26.1 ± 1.0 | 86.9 ± 0.3 | 35.6 ± 0.6 | 58.6 ± 1.0 | 27.7 ± 0.8 | 52.2 ± 0.6 | 31.2 ± 0.7 |
| G12EC | 27.3 ± 2.1 | 27.0 ± 1.4 | 27.1 ± 1.4 | 28.4 ± 1.0 | 67.2 ± 2.5 | 24.9 ± 1.1 | 76.5 ± 1.6 | 37.4 ± 0.9 | 49.0 ± 2.0 | 28.1 ± 1.0 |
| FedAvg | 85.6 ± 0.5 | 56.7 ± 1.0 | 58.7 ± 1.0 | 55.0 ± 0.4 | 77.2 ± 0.8 | 36.7 ± 0.5 | 68.6 ± 0.8 | 39.0 ± 0.6 | **74.1 ± 0.4** | 52.9 ± 1.0 |
| FedProx | - | - | - | - | - | - | - | - | - | - |
| Scaffold | 86.3 ± 0.5 | **58.8 ± 1.2** | 60.7 ± 0.5 | **57.6 ± 1.4** | 74.0 ± 0.6 | 35.8 ± 0.5 | **71.7 ± 0.4** | 42.5 ± 1.0 | 73.6 ± 0.1 | **53.8 ± 0.2** |
| FedInit | **86.4 ± 0.3** | 57.8 ± 1.9 | 60.6 ± 0.3 | 56.8 ± 1.0 | 74.2 ± 0.8 | 35.9 ± 0.5 | 71.5 ± 0.9 | 42.6 ± 2.0 | 73.7 ± 0.3 | 53.7 ± 0.5 |
| Ditto | - | - | - | - | - | - | - | - | - | - |
| FedSM | 85.9 ± 0.1 | 58.4 ± 1.4 | 58.5 ± 0.2 | 54.9 ± 0.7 | 77.3 ± 0.8 | 36.7 ± 0.2 | 68.7 ± 0.6 | 39.1 ± 1.4 | 74.2 ± 0.4 | 53.0 ± 0.7 |
| FedALA | 82.1 ± 11.2 | 47.0 ± 21.0 | 61.3 ± 30.7 | 46.5 ± 19.8 | 71.0 ± 35.5 | 30.8 ± 12.5 | 63.6 ± 31.8 | 36.7 ± 15.1 | 73.4 ± 0.8 | 51.9 ± 2.3 |
| Centralized | 85.8 ± 0.2 | 52.2 ± 0.8 | 74.8 ± 0.4 | 53.9 ± 0.8 | 85.3 ± 0.8 | 35.0 ± 0.6 | 73.5 ± 0.4 | 41.7 ± 1.7 | 81.6 ± 0.3 | 61.5 ± 0.3 |

- For **Micro-F1**, the t-test results indicate that the differences among federated learning algorithms are not statistically significant. This suggests that their performance in this real-world scenario is largely comparable and continues to lag behind centralized training.

- In contrast, the t-tests for **mAP** reveal significant differences between algorithms designed for heterogeneous data, such as FedProx and Scaffold, and simpler algorithms like FedAvg. These findings align with our earlier observations on the challenges imposed by long-tailed distributions.

To further evaluate model performance, we conducted additional experiments using Transformer-based architectures (Natarajan et al., 2020), with results presented in Table 14. Our findings reveal that Transformer-based models outperformed ResNet in specific federated learning algorithms, demonstrating their potential advantages. However, we also observed significant instability in certain scenarios. For example, in regularization-based algorithms such as FedProx and Ditto, the Transformer model frequently failed to converge. Similar instability was evident in adaptive algorithms like FedSM and FedALA, where some random seeds resulted in non-convergence. These challenges are consistent with prior findings, such as those reported in **Flamby** (du Terrail et al., 2022), which highlighted difficulties in employing advanced architectures like Transformers within federated learning frameworks.

# E FED-ECHO

## E.1 DESCRIPTION

Fed-ECHO consists of three datasets: CAMUS, ECHONET-DYNAMIC, and HMC-QU. The overview of Fed-ECHO is shown in Table 6.

**CAMUS.** This database consists of clinical exams from 500 patients, acquired at the University Hospital of St Etienne (France). All images are labeled with three areas: endocardium of the left ventricle ($LV_{Endo}$), epicardium of the left ventricle ($LV_{Epi}$), and left atrium wall (LA). The image size varies from $584 \times 354$ to $1945 \times 1181$.

**ECHONET-DYNAMIC.** This database contains 10,0230 echocardiogram videos where two frames are annotated with only $LV_{Endo}$ area. All frames are resized to $112 \times 112$.

**HMC-QU.** This database contains 109 echocardiogram videos collected at the Hamad Medical Corporation Hospital in Qatar. The frames of one cardiac cycle in each video are annotated with $LV_{Epi}$ area. The video frame size varies from $422 \times 636$ to $768 \times 1024$ while all labels are resized to $224 \times 224$.

### E.2 LICENSE AND ETHICS

Both CAMUS and HMC-QU datasets are open-access. HMC-QU database requires the user to have a Kaggle account, while the ECHONET-DYNAMIC database requires the user to have a Stanford AIMI account and to accept its agreement. It is licensed under the Stanford University Dataset Research Use Agreement.

### E.3 DOWNLOAD AND PREPROCESSING

#### E.3.1 DOWNLOAD

The three datasets can be downloaded using the URLs below:

1. **CAMUS:** `https://humanheart-project.creatis.insa-lyon.fr/database/#collection/6373703d73e9f0047faa1bc8`
2. **ECHONET-DYNAMIC:** `https://echonet.github.io/dynamic/index.html#access`
3. **HMC-QU:** `https://www.kaggle.com/datasets/aysendegerli/hmcqu-dataset/data`

#### E.3.2 PREPROCESSING

Raw echocardiograms have varying frame sizes, modalities, and mask labels, which must be standardized before training. Therefore, as a first step, we extract frames that are annotated and store them as images. We then resize them to a common ($112 \times 112$) shape. Finally, we align the labels of records in different databases. We use 1, 2, 3 representing $LV_{Endo}$, $LV_{Epi}$ and LA respectively. The samples of Fed-ECHO are shown in Figure6.

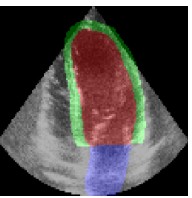 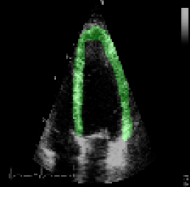

(a) Sample from Institution 1.     (b) Sample from Institution 2.     (c) ample from Institution 3.

Figure 6: Echocardiogram of each institution in Fed-ECHO. $LV_{Endo}$, $LV_{Epi}$ and LA are shown in red, green and blue respectively.

### E.4 BASELINE, LOSS FUNCTION AND EVALUATION

**Baseline Model.** A U-net architecture is employed in this study, utilizing echocardiographic images as input to forecast masks delineating four distinct cardiac regions. The U-net model represents a conventional convolutional neural network design frequently deployed in the realm of biomedical image segmentation endeavors. Its application is tailored towards semantic segmentation, a process wherein individual pixels within an image are categorized based on semantic content.

**Loss function.** We use a CrossEntropy Loss (CELoss) for training. Note that, for centralized supervised learning and client training in FedAvg, FedProx, Scaffold, and Ditto strategies, we ignore label with value 0 when calculating CELoss for data from client 2 or 3, since region with label 0 may not be true ground truth in these clients.

**Evaluation Metrics.** We use the Dice similarity index and 2D Hausdorff distance ($d_H$) to measure the accuracy of the segmentation output. Dice index is calculated as:

$$\text{DICE}(\mathbf{y}, \hat{\mathbf{y}}) = \frac{2 \sum_{i=1}^{n} y_i \hat{y}_i}{\sum_{i=1}^{n} y_i + \sum_{i=1}^{n} \hat{y}_i} \tag{4}$$

The Hausdorff distance is calculated as:

$$\text{d}_H(\mathbf{y}, \hat{\mathbf{y}}) = \max\{d(\mathbf{y}, \hat{\mathbf{y}}), d(\hat{\mathbf{y}}, \mathbf{y})\}, \tag{5}$$

where $d(\mathbf{y}, \hat{\mathbf{y}})$ represents the minimum distance among points at the edge of $\mathbf{y}$ and points at the edge of $\hat{\mathbf{y}}$.

Note that, to better measure the model segmentation performance, for clients 2, and 3, we select only 200 labeled frames for testing.

### E.5 TRAINING DETAIL

**Optimization parameters.** We optimize our model using the SGD optimizer, with a batch size of 32. We train our model for 50 epochs on one NVIDIA A100-PCIE-40GB. To ensure robustness and statistical reliability, we repeat each experiment five times and report the mean and standard deviation of the results.

**Hyperparameter Search** For centralized and local model training, we first explore learning rates from the set {1e-4, 1e-3, 1e-2, 1e-1.5, 1e-1} during centralized model training. The learning rate that achieves the best Dice index is then utilized for local model training. For the federated learning strategies, we employ the following hyperparameter grid:

- For clients' learning rates (all strategies except Fed-Consist): {1e-4, 1e-3, 1e-2, 1e-1.5, 1e-1}.
- For server size learning rate (Scaffold strategy only): {1e-2, 1e-1, 1.0}.
- For FedProx and Ditto strategies, the parameter $\mu$ is selected from {1e-2, 1e-1, 1.0}.
- For FedInit, the parameter $\beta$ is chosen from {1e-1, 1e-2, 1e-3}.
- For FedSM, the parameters $\gamma$ and $\lambda$ are set to {0, 0.1, 0.7, 0.9} and {0.1, 0.3, 0.5, 0.7, 0.9}, respectively.
- For FedALA, the parameters layer index, $\eta$, threshold, and num_per_loss are fixed at 2, 1.0, 0.1, and 10, respectively, while rand_percent is chosen from {5, 50, 80}.

For Fed-Consist, we introduce Gaussian noise with a variance of 0.1 as augmentation. The learning rates for labeled clients are searched from {1e-2, 1e-3, 1e-4}, while those for unlabeled clients are explored within {1e-3, 1e-4, 1e-5, 5e-6, 1e-6}. The parameter $\tau$ is varied from {0.5, 0.7, 0.9}.

Additionally, for FedPSL, we further search the parameters $\alpha$ and $\beta$ from {1e-0.5, 1e-1, 1e-1.5, 1e-2, 1e-3} and {1e-1, 1e-1.5, 1e-2, 1e-3, 1e-4, 1e-5}, respectively. The optimal values found are $\alpha = 1e-1.5$ and $\beta = 1e-5$.

### E.6 SUPPLEMENTARY EXPERIMENTS

We conducted pairwise t-tests on the **DICE** and **Hausdorff distance** ($d_H$) metrics, with the resulting p-values presented in Table 17. The t-test results reveal significant differences between advanced algorithms designed to address heterogeneity or semi-supervised scenarios (e.g., FedConsist) and

Table 15: Hyperparameters used for the Fed-ECHO with Unet model.

| | Fed-ECHO | | | | | | | | | |
|---|---|---|---|---|---|---|---|---|---|---|
| Methods | learning rate | optimizer | learning rate server | mu | beta | lambda | gamma | rand_percent | $\tau$ |
| Central.(sup) | 0.1 | torch.optim.SGD | - | - | - | - | - | - | - |
| Central.(ssup) | 0.1 | torch.optim.SGD | - | - | - | - | - | - | - |
| FedAvg | 0.1 | torch.optim.SGD | - | - | - | - | - | - | - |
| FedProx | 0.1 | torch.optim.SGD | - | 0.1 | - | - | - | - | - |
| Scaffold | 0.1 | torch.optim.SGD | 1.0 | - | - | - | - | - | - |
| FedInit | 0.1 | torch.optim.SGD | 1.0 | - | 1e-2 | - | - | - | - |
| Ditto | 0.1 | torch.optim.SGD | - | 0.1 | - | - | - | - | - |
| FedSM | 0.1 | torch.optim.SGD | 1.0 | - | - | 0.1 | 0 | - | - |
| FedALA | 0.1 | torch.optim.SGD | 1.0 | - | - | - | - | 5 | - |
| FedPSL | 0.1 | torch.optim.SGD | 1.0 | - | 1e-5 | - | - | - | - |
| Fed-Consist | 0.0001(labeled client) 1e-6(unlabeled client) | torch.optim.SGD | - | - | - | - | - | - | 0.9 |

Table 16: Hyperparameters used for the Fed-ECHO with Unetr model.

| | Fed-ECHO | | | | | | | | | |
|---|---|---|---|---|---|---|---|---|---|---|
| Methods | learning rate | optimizer | learning rate server | mu | beta | lambda | gamma | rand_percent | $\tau$ |
| Central.(sup) | 0.00001 | torch.optim.Adam | - | - | - | - | - | - | - |
| Central.(ssup) | 0.0003162 | torch.optim.Adam | - | - | - | - | - | - | - |
| FedAvg | 0.00003162 | torch.optim.Adam | - | - | - | - | - | - | - |
| FedProx | 0.0001 | torch.optim.Adam | - | 0.01 | - | - | - | - | - |
| Scaffold | 0.0001 | torch.optim.Adam | 1.0 | - | - | - | - | - | - |
| FedInit | 0.0001 | torch.optim.Adam | 1.0 | - | 0.001 | - | - | - | - |
| Ditto | 0.0001 | torch.optim.Adam | - | 1.0 | - | - | - | - | - |
| FedSM | 0.0001 | torch.optim.Adam | 1.0 | - | - | 0.5 | 0.1 | - | - |
| FedALA | 0.0001 | torch.optim.Adam | 1.0 | - | - | - | - | 5 | - |
| FedPSL | 0.0001 | torch.optim.Adam | 1.0 | - | 0.0001 | - | - | - | - |
| Fed-Consist | 0.00003162(labeled client) 1e-08(unlabeled client) | torch.optim.Adam | - | - | - | - | - | - | 0.9 |

Table 17: The p-values of the pairwise t-tests on the performance, DICE (left) and $d_H$ (right), of different FL algorithms on Fed-ECHO, where those with significant performance differences detected are shown in **bold**. Here, FedCon stands for Fed-Consist.

| | FedProx | Scaffold | FedInit | Ditto | FedSM | FedALA | FedCon | FedPSL |
|---|---|---|---|---|---|---|---|---|
| FedAvg | **0.02125** | **0.00047** | **0.00035** | **0.00034** | **0.00102** | 0.54492 | **0.00059** | 0.05971 |
| FedProx | - | 0.45721 | 0.42670 | 0.56817 | 0.52675 | **0.02464** | 0.84424 | **0.04052** |
| Scaffold | - | - | **0.09345** | 0.40242 | 0.33671 | **0.00002** | **0.00104** | **0.00005** |
| FedInit | - | - | - | 0.23269 | 0.23387 | **0.00002** | **0.00014** | **0.00002** |
| Ditto | - | - | - | - | 0.95874 | **0.00005** | 0.07293 | **0.00011** |
| FedSM | - | - | - | - | - | **0.00014** | 0.14410 | **0.00045** |
| FedALA | - | - | - | - | - | - | **0.00001** | 0.05358 |
| FedCon | - | - | - | - | - | - | - | **0.00003** |

| | FedProx | Scaffold | FedInit | Ditto | FedSM | FedALA | FedCon | FedPSL |
|---|---|---|---|---|---|---|---|---|
| FedAvg | **0.02125** | **0.00047** | **0.00035** | **0.00034** | **0.00102** | 0.54492 | **0.00059** | 0.05971 |
| FedProx | - | 0.45721 | 0.42670 | 0.56817 | 0.52675 | **0.02464** | 0.84424 | **0.04052** |
| Scaffold | - | - | **0.09345** | 0.40242 | 0.33671 | **0.00002** | **0.00104** | **0.00005** |
| FedInit | - | - | - | 0.23269 | 0.23387 | **0.00002** | **0.00014** | **0.00002** |
| Ditto | - | - | - | - | 0.95874 | **0.00005** | 0.07293 | **0.00011** |
| FedSM | - | - | - | - | - | **0.00014** | 0.14410 | **0.00045** |
| FedALA | - | - | - | - | - | - | **0.00001** | 0.05358 |
| FedCon | - | - | - | - | - | - | - | **0.00003** |

Table 18: The performances of different FL methods on Fed-ECHO with UNETR, with DICE and $d_H$ representing DICE index and Hausdorff distance respectively.The best results for each configuration are highlighted in **bold**, while the second-best results are underlined.

| Mthods | CAMUS | | ECHONET-DYNAMIC | | HMC-QU | | GLOBAL | |
|---|---|---|---|---|---|---|---|---|
| | Dice↑ | $d_H$↓ | Dice↑ | $d_H$↓ | Dice↑ | $d_H$↓ | Dice↑ | $d_H$↓ |
| CAMUS | $85.8 \pm 0.2$ | $7.101 \pm 0.282$ | $56.1 \pm 3.2$ | $36.151 \pm 2.698$ | $47.1 \pm 5.6$ | $39.005 \pm 8.338$ | $63.0 \pm 2.9$ | $27.419 \pm 3.268$ |
| ECHONET-DYNAMIC | $26.8 \pm 1.0$ | $70.407 \pm 0.460$ | $91.6 \pm 0.1$ | $4.745 \pm 0.079$ | - | - | $39.5 \pm 0.3$ | $58.384 \pm 0.150$ |
| HMC-QU | $17.2 \pm 0.4$ | $76.361 \pm 0.522$ | - | - | $93.5 \pm 0.3$ | $5.010 \pm 1.026$ | $36.9 \pm 0.1$ | $60.457 \pm 0.485$ |
| FedAvg | $36.8 \pm 5.7$ | $50.527 \pm 7.823$ | $50.4 \pm 2.5$ | $44.763 \pm 4.976$ | $48.9 \pm 11.1$ | $51.500 \pm 5.175$ | $45.3 \pm 5.0$ | $48.930 \pm 5.308$ |
| FedProx | $76.0 \pm 1.0$ | $15.757 \pm 1.403$ | $\mathbf{62.5 \pm 1.8}$ | $43.621 \pm 4.113$ | $\mathbf{58.1 \pm 4.5}$ | $33.886 \pm 6.959$ | $65.6 \pm 2.0$ | $31.088 \pm 3.177$ |
| Scaffold | $83.1 \pm 0.4$ | $9.741 \pm 1.395$ | $54.8 \pm 2.5$ | $37.102 \pm 1.973$ | $53.2 \pm 2.5$ | $\mathbf{24.412 \pm 4.031}$ | $63.7 \pm 0.9$ | $\mathbf{23.752 \pm 1.841}$ |
| FedInit | $82.6 \pm 0.9$ | $\underline{10.753 \pm 1.823}$ | $57.0 \pm 6.8$ | $37.314 \pm 2.862$ | $\underline{53.8 \pm 2.8}$ | $26.573 \pm 7.028$ | $64.5 \pm 2.6$ | $24.880 \pm 2.933$ |
| Ditto | $80.9 \pm 0.6$ | $12.901 \pm 0.724$ | $51.4 \pm 3.7$ | $41.215 \pm 2.257$ | $49.7 \pm 1.7$ | $37.581 \pm 6.761$ | $62.1 \pm 1.2$ | $31.022 \pm 2.428$ |
| FedSM | $63.0 \pm 15.0$ | $21.180 \pm 7.045$ | $56.7 \pm 5.7$ | $35.526 \pm 3.485$ | $53.2 \pm 5.5$ | $28.951 \pm 5.420$ | $57.6 \pm 7.8$ | $28.552 \pm 4.105$ |
| FedALA | $81.2 \pm 1.4$ | $13.900 \pm 2.548$ | $54.7 \pm 2.9$ | $\underline{34.515 \pm 1.180}$ | $36.8 \pm 6.6$ | $49.606 \pm 7.051$ | $52.3 \pm 5.6$ | $35.803 \pm 6.580$ |
| Fed-Consist | $\mathbf{86.1 \pm 0.0}$ | $\mathbf{6.848 \pm 0.020}$ | $\underline{59.9 \pm 0.0}$ | $\mathbf{30.296 \pm 0.123}$ | $51.0 \pm 0.0$ | $40.413 \pm 0.100$ | $\mathbf{65.7 \pm 0.0}$ | $25.852 \pm 0.042$ |
| FedPSL | $49.9 \pm 8.3$ | $27.877 \pm 2.148$ | $48.3 \pm 2.9$ | $37.767 \pm 2.666$ | $49.7 \pm 2.9$ | $45.388 \pm 6.942$ | $49.3 \pm 3.0$ | $37.011 \pm 2.564$ |
| Centralized(sup) | $82.9 \pm 0.5$ | $9.850 \pm 0.338$ | $52.9 \pm 2.4$ | $37.435 \pm 1.326$ | $58.9 \pm 3.1$ | $22.309 \pm 5.917$ | $64.9 \pm 1.8$ | $23.198 \pm 2.424$ |
| Centralized(ssup) | $87.0 \pm 0.5$ | $6.315 \pm 0.268$ | $91.2 \pm 0.2$ | $4.999 \pm 0.151$ | $90.8 \pm 0.6$ | $3.066 \pm 0.170$ | $89.7 \pm 0.3$ | $4.794 \pm 0.093$ |

simpler methods like FedAvg. These findings further support our conclusion that specialized algorithmic strategies consistently outperform baseline methods under challenging non-IID conditions.

We also conducted experiments using **Unetr** (Hatamizadeh et al., 2022), a Transformer-based model, to further validate our findings. The results are shown in Figure 18 While **Unetr** demonstrated some performance variations compared to **U-Net**, the overall conclusions of our study remained robust. Specifically, federated learning algorithms designed to tackle challenges such as data heterogeneity (e.g., non-IID data, long-tail distributions, and label incompleteness) and semi-supervised learning algorithms consistently outperformed simpler baseline methods, regardless of the underlying model architecture.

