# OpenReview forum: "FedCVD: The First Real-World Federated Learning Benchmark on Cardiovascular Disease Data"
_ICLR.cc/2025/Conference — Submitted to ICLR 2025_

### Official Review · Reviewer_dwEh · 2024-10-31

**Soundness:** 3
**Presentation:** 2
**Contribution:** 3
**Rating:** 6
**Confidence:** 3

**Summary:**

This work collates and harmonises 7 open-access cardiology datasets (4 ECG, 3 ECHO) to create a realistic benchmark for federated learning. The authors show that existing federated learning benchmarks may not adequately reflect real-world challenges in cardiovascular research: 1) natural partitions are needed to reflect inter-institution heterogeneity; 2) clinical outcomes of interest are often not binary but multi-class and long-tailed; 3) labelling may differ significantly between

**Strengths:**

Medical machine learning still lags behind many other domains, in part owing to the difficulties of collecting sufficiently large and diverse data. Federated learning is increasingly important to address this challenge and allow the training of larger models in secure and privacy-preserving manner. However, studies that evaluate the performance of different FL algorithms rely on artificially federated data that does not reflect conditions encountered in the wild. This work introduces two complex datasets that can help facilitate more realistic benchmarking of FL algorithms.

**Weaknesses:**

The paper tries to do many things at once, which makes it difficult to read at times. Here are some examples:
- Scope: while the work presents itself as a benchmark for CVD, in terms of modalities it contains separate benchmarks for a time series (ECG) and 2D images/videos (ECHO). The benchmarks in Table 1 span additional modalities including tabular data (FedTD), time-to-event data (Flamby), graphs (FedDTI), NLP (FEDLEGAL), and multimodal data (FedMultimodal). It is unclear whether they are even comparable.
- Related work: the related work section does not have a clear logical flow. As far as I can see, the main arguments are: 1) AI is important in cardiology and spans ECGs and images among other modalities (could be argued more concisely); 2) most existing applications are trained on single centre data but multicentre data is needed for reasons XYZ (currently not clear why multicentre data is needed); 3) federated learning provides a privacy-preserving way to share data between institutions but most studies to date only show proof-of-concept with artificially federated data; 4) realistic benchmarks for FL are needed (why?) and some work has been done in that direction, but they do not completely consider the challenges faced in FL for CVD. Section headings may help to guide the reader through this section.
- The Proposed FedCVD: This section introduces the datasets, metrics, baseline models, and challenges. However, it does not actually introduce the framework. Here I would expect to see the flow of an experiment, how the framework/API supports adding new datasets or models, etc. The metrics and baseline models, on the other hand, belong into the experiment section and are indeed repeated there.
- Experiments: The current version is structured by dataset (Fed-ECG and Fed-ECHO). It might make it more readable if the authors group the results by challenges (non-IID, long-tail, incomplete) instead. The experiment section also mixes methods with results. For example, on page 8 F1-STD and Top-K are introduced right next to the discussion of the results. Consider explaining your metrics earlier in the experiment details section.

**Questions:**

Major suggestions:
- Variability in the algorithms' performance estimates seemed quite substantial at times (e.g., Table 2 GLOBAL Mi-F1 FedAvg 67.9+/-3.8 compared to the leading Scaffold 70.1+/-0.8) , making it unclear whether the differences between algorithms are actually meaningful or just due to chance. I was unable to find information on how variability was calculated. Please include that information and consider adding formal tests for differences.
- It is not immediately obvious to me how some algorithms like Ditto in Table 2 can have a much lower Mi-F1 in the GLOBAL set (68.1) than in any single local set (min 73.4). Do they authors have an intuitive explanation for this?
- I was unable to find any details on the semi-supervised algorithms for the ECHO data. Please add a clear description of what was done in those cases.

Minor suggestions:
- it might be worth replacing Client 1-4 with the actual dataset names to make sure which local performance is shown.

---

> ### Author Response · Authors · 2024-11-27
> **Responses to Reviewer dwEh (1/3)**
>
> Thank you for your valuable and detailed feedback. We are open to discussion with the reviewer regarding these points as well as the revised paper.
>
> We have revised the **Related Work** section by incorporating section headings to enhance the logical flow and provide clearer guidance for readers ***(Line 43-45, 148, 162-164, 177, 183-184)***. Additionally, we have restructured the **Proposed FedCVD** and **Experiments** sections ***(Line 311-332, 354-358, 410-415)***, relocating the descriptions of baseline models and metrics to earlier subsections. This adjustment ensures better organization and improves the accessibility of the content for readers.
>
> ## Q1: Variability in the algorithms' performance estimates seemed quite substantial at times , making it unclear whether the differences between algorithms are actually meaningful or just due to chance.  Please include that information and consider adding formal tests for differences.
>
> Thank you for bringing up this important concern regarding the variability and significance of performance differences.
>
> To address this, we conducted experiments using five random seeds and reported the mean and standard deviation of the results. Additionally, we performed t-tests to assess the statistical significance of the observed differences in various metrics.
>
> - The p-values of the pairwise t-tests on the performance Micro-F1 of different FL algorithms on Fed-ECG.
>
> |          | FedProx | Scaffold | FedInit | Ditto | FedSM | FedALA |
> | -------- | ------- | -------- | ------- | ----- | ----- | ------ |
> | FedAvg   | 0.73    | 0.27     | 0.94    | 0.93  | 0.71  | 0.99   |
> | FedProx  |         | 0.26     | 0.74    | 0.74  | 0.97  | 0.59   |
> | Scaffold |         |          | 0.23    | 0.29  | 0.42  | 0.12   |
> | FedInit  |         |          |         | 0.98  | 0.63  | 0.91   |
> | Ditto    |         |          |         |       | 0.53  | 0.88   |
> | FedSM    |         |          |         |       |       | 0.63   |
>
> - The p-values of the pairwise t-tests on the performance mAP of different FL algorithms on Fed-ECG.
>
> |          | FedProx   | Scaffold  | FedInit | Ditto     | FedSM     | FedALA    |
> | -------- | --------- | --------- | ------- | --------- | --------- | --------- |
> | FedAvg   | **0.009** | **0.016** | 0.201   | **0.048** | 0.462     | 0.983     |
> | FedProx  |           | 0.704     | 0.373   | **0.011** | 0.063     | 0.056     |
> | Scaffold |           |           | 0.452   | **0.005** | 0.069     | **0.017** |
> | FedInit  |           |           |         | 0.058     | 0.646     | 0.531     |
> | Ditto    |           |           |         |           | **0.035** | **0.026** |
> | FedSM    |           |           |         |           |           | 0.535     |

---

> ### Author Response · Authors · 2024-11-27
> **Responses to Reviewer dwEh (2/3)**
>
> - The p-values of the pairwise t-tests on the performance DICE of different FL algorithms on Fed-ECHO.
>
> |            | fedprox     | scaffold    | fedinit     | ditto       | fedsm       | fedala      | fedconsist  | fedpsl      |
> | ---------- | ----------- | ----------- | ----------- | ----------- | ----------- | ----------- | ----------- | ----------- |
> | fedavg     | **0.02125** | **0.00047** | **0.00035** | **0.00034** | **0.00102** | 0.54492     | **0.00059** | 0.05971     |
> | fedprox    |             | 0.45721     | 0.42670     | 0.56817     | 0.52675     | **0.02464** | 0.84424     | **0.04052** |
> | scaffold   |             |             | **0.09345** | 0.40242     | 0.33671     | **0.00002** | **0.00104** | **0.00005** |
> | fedinit    |             |             |             | 0.23269     | 0.23387     | **0.00002** | **0.00014** | **0.00002** |
> | ditto      |             |             |             |             | 0.95874     | **0.00005** | 0.07293     | **0.00011** |
> | fedsm      |             |             |             |             |             | **0.00014** | 0.14410     | **0.00045** |
> | fedala     |             |             |             |             |             |             | **0.00001** | 0.05358     |
> | fedconsist |             |             |             |             |             |             |             | **0.00003** |
>
> - The p-values of the pairwise t-tests on the performance $d_H$ of different FL algorithms on Fed-ECHO.
>
> |            | fedprox     | scaffold    | fedinit     | ditto       | fedsm       | fedala      | fedconsist  | fedpsl      |
> | ---------- | ----------- | ----------- | ----------- | ----------- | ----------- | ----------- | ----------- | ----------- |
> | fedavg     | **0.02445** | **0.00304** | **0.00198** | **0.00086** | **0.00188** | 0.72225     | **0.03071** | 0.91994     |
> | fedprox    |             | 0.35636     | 0.29121     | 0.47471     | 0.50430     | **0.01761** | 0.07313     | **0.00631** |
> | scaffold   |             |             | 0.21219     | 0.73557     | 0.35645     | **0.00006** | **0.00011** | **0.00030** |
> | fedinit    |             |             |             | 0.33092     | 0.08627     | **0.00005** | **0.00001** | **0.00011** |
> | ditto      |             |             |             |             | 0.50695     | **0.00017** | **0.00054** | **0.00002** |
> | fedsm      |             |             |             |             |             | **0.00049** | **0.00016** | **0.00001** |
> | fedala     |             |             |             |             |             |             | **0.00238** | 0.66408     |
> | fedconsist |             |             |             |             |             |             |             | **0.00138** |
>
>
> For the **Fed-ECG** dataset:
> - Based on the t-test results for **Micro-F1**, the p-values suggest that differences among federated learning algorithms are not statistically significant, indicating that their performance remains generally far from centralized training in this real-world setting.
> - For **mAP**, the t-tests reveal that algorithms designed for heterogeneity, such as FedProx and Scaffold, exhibit significant differences compared to simpler algorithms like FedAvg. This aligns with our findings on the challenges posed by long-tail distributions.
>
> For the **Fed-ECHO** dataset:
> - The t-tests for **DICE** and **Hausdorff distance** (d_H) metrics demonstrate significant differences between advanced algorithms targeting heterogeneity or semi-supervised scenarios (e.g., FedConsist) and simpler methods such as FedAvg. These results reinforce our conclusion that tailored algorithmic approaches perform better under challenging non-IID conditions.
>
> We have included the t-test results and their detailed explanations in the appendix to provide clarity and support for the observed trends ***(Line 1178-1079, 1135-1166, 1293-1295, 1339-1347)***. Thank you for suggesting this improvement, as it enhances the robustness and credibility of our analysis.

---

> ### Author Response · Authors · 2024-11-27
> **Responses to Reviewer dwEh (3/3)**
>
> ## Q2: It is not immediately obvious to me how some algorithms like Ditto in Table 2 can have a much lower Mi-F1 in the GLOBAL set (68.1) than in any single local set (min 73.4). Do they authors have an intuitive explanation for this?
>
> Thank you for raising this important question. The observed discrepancy in Ditto’s performance **can be attributed to the nature of personalized federated learning (FL) algorithms**. In Ditto, each client maintains both a personalized local model and a global model for aggregation. The performance on the LOCAL sets is measured using the personalized models, which are optimized to suit each client’s specific data distribution. In contrast, the GLOBAL evaluation is conducted using the aggregated global model obtained from the server.
>
> Since the global model is designed to converge towards a general solution that balances the diverse objectives of all clients, its performance may degrade when evaluated against individual client distributions.
>
> ## Q3: I was unable to find any details on the semi-supervised algorithms for the ECHO data. Please add a clear description of what was done in those cases.
>
> Thank you for raising this point. For semi-supervised learning on the ECHO data, we employed a pseudo-labeling approach inspired by prior work [1]. Specifically, we first trained the model exclusively on the labeled data for 10 rounds. Subsequently, pseudo-labels were generated for the unlabeled data, which were then incorporated into the training process for an additional 40 rounds. During this phase, we dynamically adjusted the weight of the loss function for the unlabeled data using a parameter $\alpha$. This method ensured a gradual and adaptive integration of pseudo-labeled data into the training pipeline.
>
> The detailed descriptions of all methods have been incorporated into the manuscript to provide a clearer understanding of the semi-supervised algorithms and their mechanisms in the context of federated learning ***(Line 850-889)***. Thank you for your feedback!
>
>
> ## Q4: it might be worth replacing Client 1-4 with the actual dataset names to make sure which local performance is shown.
>
> Thank you for the suggestion. We have replaced the generic labels “Client 1-4” with the actual dataset names in the manuscript to clearly indicate which local performance corresponds to each dataset ***(Line 378-391, 487-499)***. This change ensures greater transparency and makes the results easier to interpret. We appreciate your feedback in helping us improve the clarity of our presentation.
>
> ## Reference
> [1] Lee, Dong-Hyun. "Pseudo-label: The simple and efficient semi-supervised learning method for deep neural networks." Workshop on challenges in representation learning, ICML. Vol. 3. No. 2. 2013.

---

### Official Review · Reviewer_sEAh · 2024-11-03

**Soundness:** 4
**Presentation:** 4
**Contribution:** 2
**Rating:** 3
**Confidence:** 5

**Summary:**

This paper proposes FedCVD, the first real-world FL benchmark for CVD diagnosis. By utilizing FL, it aims to enhance the accuracy of CVD diagnosis while preserving data privacy, enabling collaborative model training across multiple institutions. Traditional FL research has largely relied on simulated data, often failing to capture the complexity of real-world data distributions. To address this limitation, FedCVD incorporates two primary tasks, ECG classification and ECHO segmentation, based on real ECG and ECHO data collected from seven institutions, reflecting the challenges of federated learning in a real-world healthcare setting.

**Strengths:**

The primary strength of this paper is its systematic approach to evaluating CVD-related tasks using federated learning. It effectively addresses the practical challenges of Non-IID and long-tail distributions in healthcare data, which are critical issues for achieving robust model performance. Non-IID data distribution, particularly when labels are collected independently from multiple institutions rather than through a consortium, introduces complexities in model evaluation. The paper adeptly tackles this challenge, offering a realistic evaluation approach that reflects the heterogeneous data across different institutions, thus enhancing the credibility of the proposed FL setting.

Moreover, the long-tail challenge is approached innovatively using a Top-K metric, which allows for detailed performance evaluation of model accuracy across classes with varying sample distributions. This approach provides valuable insights into the model’s handling of label imbalance in real CVD data. Additionally, FedCVD rigorously analyzes various FL methodologies, demonstrating how each method performs under the specific characteristics of CVD data. With open-source code available, it contributes a practical and reproducible resource for FL research in healthcare.

**Weaknesses:**

A limitation of this paper is that, in cases of label-non-IID, the increased label sparsity across institutions may pose challenges to quantitative evaluation. As the label distribution varies more widely across institutions, certain labels may not be adequately represented, potentially compromising evaluation reliability.

Another significant limitation is whether the domain of this study aligns with the conference scope. Despite the academic value of the proposed framework, it may not be fully aligned with the themes of the conference, potentially impacting its acceptance and reception.

**Questions:**

Could you please clarify what aspects make this benchmark-focused paper suitable for this conference, particularly in terms of its contributions to evaluating real-world healthcare data?

---

> ### Author Response · Authors · 2024-11-27
> **Responses to Reviewer sEAh**
>
> Thank you for your valuable and detailed feedback. We are open to discussion with the reviewer regarding these points as well as the revised paper.
> ## Q1: A limitation of this paper is that, in cases of label-non-IID, the increased label sparsity across institutions may pose challenges to quantitative evaluation. As the label distribution varies more widely across institutions, certain labels may not be adequately represented, potentially compromising evaluation reliability.
>
>
> In prior studies addressing federated learning under long-tail label distributions, overall metrics such as accuracy are commonly used to evaluate the global performance of algorithms[1, 2]. **To maintain consistency with these established methodologies**, we have similarly adopted aggregate evaluation metrics.
>
> However, we acknowledge that these overall metrics might not fully capture the nuances of label-specific performance, particularly in scenarios with extreme label sparsity across institutions. To address this, **we have supplemented our evaluation by including label-specific performance metrics**. These additional results provide a more granular perspective on how federated learning algorithms perform across diverse labels and help mitigate concerns about evaluation reliability in label-non-IID settings.
>
> We have updated the manuscript to include these label-wise performance metrics and discussions to ensure comprehensive evaluation and transparency ***(Line 1122-1133)***. Thank you for highlighting this area for improvement.
>
> ## Q2: Another significant limitation is whether the domain of this study aligns with the conference scope. Despite the academic value of the proposed framework, it may not be fully aligned with the themes of the conference, potentially impacting its acceptance and reception. Could you please clarify what aspects make this benchmark-focused paper suitable for this conference, particularly in terms of its contributions to evaluating real-world healthcare data?
>
> Thank you for your valuable feedback. We understand your concern about the potential narrowness of focusing on a CVD benchmark. However, we would like to emphasize that the CVD domain we focus on is **a prominent area with significant potential for FL applications**. This CVD benchmark serves as a key example for FL research (on medical domains), helping to bridge the gap between theoretical developments and real-world multi-center clinical studies. To further illustrate the generalizability of FedCVD, we would like to emphasize the following points:
>
> (1) The tasks considered in our benchmark—electrocardiogram classification and echocardiogram semantic segmentation—are two **typical tasks** within the CVD domain. Furthermore, common clinical-data-based studies, can also be framed as classification or segmentation problems, underscoring the representativeness of our selected tasks.
>
> (2) The benchmark, while limited to two tasks on CVD domain, are **sufficient to cover three key challenges** central to most FL research: non-IID data, long-tail distributions, and label incompleteness. Notably, instead of synthetically simulating these challenges, Fed-ECG and Fed-ECHO offer a natural benchmark with real-world-derived data and settings.
>
> (3) The CVD-based tasks **are insightful for other medical fields**: Multi-label classification is a common task in multiple diseases diagnosis [3], and 2D semantic segmentation is a basic and important technique to aid quicker prevention and treatment of a variety of diseases. [4]. Consequently, the tasks presented can also hold potential interest for researchers in other fields where FL can be applied.
>
> ## References
> [1] Shuai, Xian, et al. "BalanceFL: Addressing class imbalance in long-tail federated learning." 2022 21st ACM/IEEE International Conference on Information Processing in Sensor Networks (IPSN). IEEE, 2022.
>
> [2] Shang, Xinyi, et al. "Fedic: Federated learning on non-iid and long-tailed data via calibrated distillation." 2022 IEEE International Conference on Multimedia and Expo (ICME). IEEE, 2022.
>
> [3] Maksoud, Eman A. Abdel, Sherif Barakat, and Mohammed Elmogy. "Medical images analysis based on multilabel classification." Machine Learning in Bio-Signal Analysis and Diagnostic Imaging. Academic Press, 2019. 209-245.
>
> [4] Salpea, Natalia, Paraskevi Tzouveli, and Dimitrios Kollias. "Medical image segmentation: A review of modern architectures." European Conference on Computer Vision. Cham: Springer Nature Switzerland, 2022.

---

### Official Review · Reviewer_b4L1 · 2024-11-03

**Soundness:** 3
**Presentation:** 3
**Contribution:** 2
**Rating:** 5
**Confidence:** 4

**Summary:**

This is a benchmark study aiming to evaluate the performance of the commonly used Federated Learning strategies for both the ECG-based arrhythmia classification task and also the echocardiogram video segmentation task. The study specifically focused to investigate the impact from non-IID in data from different sites, the long-tail distribution (where distributions among labels are very imbalanced) and the label incompleteness.

Four ECG datasets and three echo datasets were included for the benchmarking exercise. Seven federated learning algorithms were included, and their performances were compared with the one from a centralized scenario and the ones from individual sites.

**Strengths:**

1/ The manuscript was reasonably written. Most of the key concepts were explained clearly.

2/ Although there is some related work to evaluate the performance federated learning algorithms over multiple ECG datasets, this can be the first benchmarking exercise that have conducted  detailed and extensive experiments to evaluate seven FL algorithms over both the ECG-based arrhythmia classification task and also the ECHO video segmentation task.

3/ Major datasets for both tasks that are publicly available are included.

**Weaknesses:**

1/ As the authors also mentioned, the reported study is closely related to the Goto et al 2022 (https://www.ahajournals.org/doi/10.1161/CIRCULATIONAHA.121.058696) study. Although the author did mentioned about that study in the manuscript, think a more detailed explanations on how different and the new contributions of this work compared to the Goto study is still needed. Would suggest the authors to clearly compared the datasets included in both studies, the FL algorithms included, the different evaluation metrics measured, and this work's unique emphasis on non-IID in data, the long-tail distribution and the label incompleteness.

2/ For the ECG-based arrhythmia classification study, ResNet was chosen to be the base model. However, the authors did not explain the rational behind why model was chosen to be the only ONE base model for this study. Simply citing a previous paper won't justify that. Could explain what was the rational in choosing the current model?

More importantly, to report a more robust benchmarking study, suggest the author may investigate how different base models may or may not impact the performance of the included FL algorithms over the benchmarking datasets. It will useful to investigate why there may or may not be an impact?

Therefore, suggest the authors to include other base models too. Believe some customized LSTM and Transformers have been developed for ECG classification as well.  Would it be possible to include them as base models too?

3/ Similar to point 2, 2-D Unet was chosen to be the only base model in this study. Again, could explain the rationale behind picking the current base model?

Additionally, it will be also interesting to study how different segmentation base models may have different strengths in addressing the non-IID in data, the long-tail distribution and the label incompleteness challenges.

Therefore, suggest the authors to consider including one or two more base models?

Some of these possibilities could be:
Leclerc, Sarah, et al. "Deep learning for segmentation using an open large-scale dataset in 2D echocardiography." IEEE transactions on medical imaging 38.9 (2019): 2198-2210.

Wu, Huisi, et al. "Super-efficient echocardiography video segmentation via proxy-and kernel-based semi-supervised learning." Proceedings of the AAAI Conference on Artificial Intelligence. Vol. 37. No. 3. 2023.

Yang, Jiewen, et al. "GraphEcho: Graph-Driven Unsupervised Domain Adaptation for Echocardiogram Video Segmentation." Proceedings of the IEEE/CVF International Conference on Computer Vision. 2023.

and 3D models e.g.

Hatamizadeh, Ali, et al. "Unetr: Transformers for 3d medical image segmentation." Proceedings of the IEEE/CVF winter conference on applications of computer vision. 2022.

4/ For ECG datasets, would it make sense to include the Icentia11K dataset too? Although it was build mainly for self-supervised algorithms, part of the datasets do have labels? Additionally, it helps to expand the study's technical coverage if the team can expand their work to investigate how the FL learning mechanism can be combined with the newly proposed self-supervised learning models over CVD datasets.

Tan, Shawn, et al. "Icentia11k: An unsupervised representation learning dataset for arrhythmia subtype discovery." arXiv preprint arXiv:1910.09570 (2019).

**Questions:**

Please refer to the weakness section

---

> ### Author Response · Authors · 2024-11-27
> **Responses to Reviewer b4L1 (1/4)**
>
> Thank you for your valuable and detailed feedback. We are open to discussion with the reviewer regarding these points as well as the revised paper.
> ## Q1: As the authors also mentioned, the reported study is closely related to the Goto et al 2022 (https://www.ahajournals.org/doi/10.1161/CIRCULATIONAHA.121.058696) study. Although the author did mentioned about that study in the manuscript, think a more detailed explanations on how different and the new contributions of this work compared to the Goto study is still needed. Would suggest the authors to clearly compared the datasets included in both studies, the FL algorithms included, the different evaluation metrics measured, and this work's unique emphasis on non-IID in data, the long-tail distribution and the label incompleteness.
>
> Thanks for the insightful question. Below, we outline the key distinctions and potential contributions of our benchmark compared to the work by Goto et al. (2022), as well as how our emphasis on long-tail distribution and label incompleteness contributes to advancing the field.
>
> |            | FedHCD                              | FedCVD                                                               |
> | ---------- | ----------------------------------- | -------------------------------------------------------------------- |
> | Algorithms | 1 (FedAvg)                          | 9(FedAvg, FedProx, Scaffold, FedInit, ...)                           |
> | Tasks      | Binary Classification               | Multi-label Classification & Segmentation                            |
> | Metrics    | AUROC                               | Micro-F1, Macro-F1, mean average precision, DICE, Hausdorff distance |
> | Challenges | Non-IID(demographics heterogeneity) | Non-IID, Long-tail distribution, Label incompleteness                |
> | Datasets   | 3 private datasets                  | 7 public datasets                                                    |
>
> **More FL Algorithms.** As summarized in the table, Goto et al.’s work primarily applies *a single FL algorithm*, FedAvg, to a binary classification task focused on hypertrophic cardiomyopathy detection. This limits its scope to exploring only a narrow aspect of FL in the context of cardiovascular disease (CVD). In contrast, our benchmark investigates a broader range of FL algorithms (***nine*** in total, including FedAvg, FedProx, Scaffold, and so on) across more diverse and complex tasks, such as multi-label classification and 2D segmentation.
>
>
> **More Challenging Scenarios**. Furthermore, while Goto et al.’s work considers Non-IID challenges stemming from demographic heterogeneity, our benchmark incorporates additional, highly relevant challenges in FL scenarios, namely *long-tail distribution* and *label incompleteness*. Addressing both challenges is particularly crucial in real-world FL applications, including healthcare. Long-tail distribution arises naturally in fine-grained medical tasks, where some institutions may only possess a small number of rare disease samples. Existing FL methods often fail to preserve knowledge from such rare but critical classes, leading to suboptimal performance. By contrast, our benchmark highlights this issue, enabling the development and evaluation of algorithms that better capture these underrepresented cases. Similarly, label incompleteness is prevalent in federated healthcare due to varying institutional priorities and capacities, where different institutions annotate overlapping but non-identical subsets of labels. Designing methods to effectively leverage partially annotated data can significantly enhance model generalization and utility across all participating parties.
>
> To the best of our knowledge, prior benchmarks, including Goto et al., have not thoroughly considered these challenges, which are critical for improving the robustness and applicability of FL in real-world medical contexts. By incorporating long-tail distribution and label incompleteness into our benchmark, we aim to provide the community with a more rigorous and realistic framework for evaluating FL approaches in healthcare, paving the way for future innovations.

---

> ### Author Response · Authors · 2024-11-27
> **Responses to Reviewer b4L1 (2/4)**
>
> ## Q2: For the ECG-based arrhythmia classification study, ResNet was chosen to be the base model. However, the authors did not explain the rational behind why model was chosen to be the only ONE base model for this study. Simply citing a previous paper won't justify that. Could explain what was the rational in choosing the current model? More importantly, to report a more robust benchmarking study, suggest the author may investigate how different base models may or may not impact the performance of the included FL algorithms over the benchmarking datasets. It will useful to investigate why there may or may not be an impact?Therefore, suggest the authors to include other base models too. Believe some customized LSTM and Transformers have been developed for ECG classification as well. Would it be possible to include them as base models too?
>
>
> We selected ResNet as the base model due to its **widespread use** in prior ECG-based arrhythmia classification studies and its **consistent performance advantages over alternative models**[1, 2] ***(Line 354-358)***. Its proven reliability made it a suitable choice for the initial benchmark.
>
> In addition to ResNet, **we incorporated a Transformer-based model[3] into our study** to investigate the impact of model architecture on federated learning performance. Transformers are gaining popularity in ECG-related tasks for their ability to capture long-range dependencies effectively.
>
> Their Performance (mAP% $\uparrow$) on Fed-ECG are as follows:
>
> | Methods  |        SPH         |       PTB-XL       |        SXPH        |       G12EC        |       GLOBAL       |
> | -------- | :----------------: | :----------------: | :----------------: | :----------------: | :----------------: |
> | SPH      |   52.8 $\pm$ 3.3   |   36.6 $\pm$ 0.6   |   20.7 $\pm$ 0.1   |   27.5 $\pm$ 1.8   |   32.9 $\pm$ 0.9   |
> | PTB-XL   |   35.0 $\pm$ 0.9   |   48.4 $\pm$ 1.4   |   18.5 $\pm$ 1.5   |   26.5 $\pm$ 0.5   |   31.2 $\pm$ 1.6   |
> | SXPH     |   26.7 $\pm$ 1.6   |   26.1 $\pm$ 1.0   |   35.6 $\pm$ 0.6   |   27.7 $\pm$ 0.8   |   31.2 $\pm$ 0.7   |
> | G12EC    |   27.0 $\pm$ 1.4   |   28.4 $\pm$ 1.0   |   24.9 $\pm$ 1.1   |   37.4 $\pm$ 0.9   |   28.1 $\pm$ 1.0   |
> | FedAvg   |   56.7 $\pm$ 1.0   |   55.0 $\pm$ 0.4   |   36.7 $\pm$ 0.5   |   39.0 $\pm$ 0.6   |   52.9 $\pm$ 1.0   |
> | FedProx  |         -          |         -          |         -          |         -          |         -          |
> | Scaffold | **58.8 $\pm$ 1.2** | **57.6 $\pm$ 1.4** |   35.8 $\pm$ 0.5   |   42.5 $\pm$ 1.0   | **53.8 $\pm$ 0.2** |
> | FedInit  |   57.8 $\pm$ 1.9   |   56.8 $\pm$ 1.0   |   35.9 $\pm$ 0.5   | **42.6 $\pm$ 2.0** |   53.7 $\pm$ 0.5   |
> | Ditto    |         -          |         -          |         -          |         -          |         -          |
> | FedSM    |   58.4 $\pm$ 1.4   |   54.9 $\pm$ 0.7   | **36.7 $\pm$ 0.2** |   39.1 $\pm$ 1.4   |   53.0 $\pm$ 0.7   |
> | FedALA   |  47.0 $\pm$ 21.0   |  46.5 $\pm$ 19.8   |  30.8 $\pm$ 12.5   |  36.7 $\pm$ 15.1   |   51.9 $\pm$ 2.3   |
> | Central. |   52.2 $\pm$ 0.8   |   53.9 $\pm$ 0.8   |   35.0 $\pm$ 0.6   |   41.7 $\pm$ 1.7   |   61.5 $\pm$ 0.3   |
>
>
> Our findings reveal that Transformer-based models outperformed ResNet in certain federated learning algorithms, showcasing their potential advantages.
>
> However, we also observed **significant instability of Transformers** in specific scenarios. For instance, in regularization-based algorithms such as FedProx and Ditto, the Transformer model frequently failed to converge. Similar instability was noted in adaptive algorithms like FedSM and FedALA, where some random seeds led to non-convergence. These challenges align with observations in works such as *Flamby*[4], which also reported difficulties in leveraging advanced architectures like Transformers in federated settings.
>
> We have also documented these findings and discussions in the Appendix ***(Line 1144-1175)***. Thank you for helping us broaden the scope and insight of this benchmarking study.

---

> ### Author Response · Authors · 2024-11-27
> **Responses to Reviewer b4L1 (3/4)**
>
> ## Q3: Similar to point 2, 2-D Unet was chosen to be the only base model in this study. Again, could explain the rationale behind picking the current base model?Additionally, it will be also interesting to study how different segmentation base models may have different strengths in addressing the non-IID in data, the long-tail distribution and the label incompleteness challenges. Therefore, suggest the authors to consider including one or two more base models?
>
> We selected 2-D U-Net as the base model because it has been **widely adopted in prior echocardiogram segmentation studies and has consistently demonstrated superior performance** in various tasks[5] ***(Line 354-358)***. Its robustness and extensive validation in the literature made it a natural choice for our benchmark.
>
> To address your suggestion, **we expanded our study by including Unetr**[6], a Transformer-based model for medical image segmentation. Unetr leverages the power of Transformers to capture both local and global context, which is particularly valuable for handling complex segmentation tasks.
>
> Our experiments showed that while Unetr exhibited some performance differences compared to U-Net, **the overall conclusions of our study remained consistent**. Specifically, federated learning algorithms designed to address data heterogeneity (e.g., non-IID data, long-tail distribution, and label incompleteness) and semi-supervised learning algorithms consistently outperformed simpler baseline approaches, regardless of the base model used.
>
> We have included the experimental results and hyperparameter details for Unetr in the appendix to provide additional insights ***(Line 1350-1374)***. This expansion reinforces the robustness of our findings and underscores the importance of algorithm design in federated learning. Thank you for suggesting this improvement, which has enriched the depth of our analysis.
>
>
> Their Performance (DICE% $\uparrow$) on Fed-ECHO are as follows:
>
> | Methods         |       CAMUS        |  ECHONET-DYNAMIC   |       HMC-QU       |       GLOBAL       |
> | --------------- | :----------------: | :----------------: | :----------------: | :----------------: |
> | CAMUS           |   85.8 $\pm$ 0.2   |   56.1 $\pm$ 3.2   |   47.1 $\pm$ 5.6   |   63.0 $\pm$ 2.9   |
> | ECHONET-DYNAMIC |   26.8 $\pm$ 1.0   |   91.6 $\pm$ 0.1   |         -          |   39.5 $\pm$ 0.3   |
> | HMC-QU          |   17.2 $\pm$ 0.4   |         -          |   93.5 $\pm$ 0.3   |   36.9 $\pm$ 0.1   |
> | FedAvg          |   36.8 $\pm$ 5.7   |   50.4 $\pm$ 2.5   |  48.9 $\pm$ 11.1   |   45.3 $\pm$ 5.0   |
> | FedProx         |   76.0 $\pm$ 1.0   | **62.5 $\pm$ 1.8** |   58.1 $\pm$ 4.5   |   65.6 $\pm$ 2.0   |
> | Scaffold        |   83.1 $\pm$ 0.4   |   54.8 $\pm$ 2.5   |   53.2 $\pm$ 2.5   |   63.7 $\pm$ 0.9   |
> | FedInit         |   82.6 $\pm$ 0.9   |   57.0 $\pm$ 6.8   | **53.8 $\pm$ 2.8** |   64.5 $\pm$ 2.6   |
> | Ditto           |   80.9 $\pm$ 0.6   |   51.4 $\pm$ 3.7   |   49.7 $\pm$ 1.7   |   62.1 $\pm$ 1.2   |
> | FedSM           |  63.0 $\pm$ 15.0   |   56.7 $\pm$ 5.7   |   53.2 $\pm$ 5.5   |   57.6 $\pm$ 7.8   |
> | FedALA          |   81.2 $\pm$ 1.4   |   54.7 $\pm$ 2.9   |   36.8 $\pm$ 6.6   |   52.3 $\pm$ 5.6   |
> | Fed-Consit      | **86.1 $\pm$ 0.0** |   59.9 $\pm$ 0.0   |   51.0 $\pm$ 0.0   | **65.7 $\pm$ 0.0** |
> | FedPSL          |   49.9 $\pm$ 8.3   |   48.3 $\pm$ 2.9   |   49.7 $\pm$ 2.9   |   49.3 $\pm$ 3.0   |
> | Central.(sup)   |   82.9 $\pm$ 0.5   |   52.9 $\pm$ 2.4   |   58.9 $\pm$ 3.1   |   64.9 $\pm$ 1.8   |
> | Central.(ssup)  |   87.0 $\pm$ 0.5   |   91.2 $\pm$ 0.2   |   90.8 $\pm$ 0.6   |   89.7 $\pm$ 0.3   |

---

> ### Author Response · Authors · 2024-11-27
> **Responses to Reviewer b4L1 (4/4)**
>
> ## Q4: For ECG datasets, would it make sense to include the Icentia11K dataset too? Although it was build mainly for self-supervised algorithms, part of the datasets do have labels? Additionally, it helps to expand the study's technical coverage if the team can expand their work to investigate how the FL learning mechanism can be combined with the newly proposed self-supervised learning models over CVD datasets.
>
> As suggested, we carefully examined the Icentia11K[7] dataset, which is an excellent resource for semi-supervised ECG model training. However, we find that its 1-lead ECG data  differs significantly from the 10s 12-lead ECG datasets we have adopted. This discrepancy **creates a substantial challenge due to the incomplete feature space**, which complicates the application of mainstream federated learning algorithms that assume a consistent feature space across clients.
>
> Nevertheless, our framework is highly flexible, allowing us to incorporate the Icentia11K dataset into the FedCVD benchmark easily. We plan to explore this further in future work with additional experiments and analyses.
>
> ## Reference
> [1] Strodthoff, Nils, et al. "Deep learning for ECG analysis: Benchmarks and insights from PTB-XL." IEEE journal of biomedical and health informatics 25.5 (2020): 1519-1528.
>
> [2] Ribeiro, Antônio H., et al. "Automatic diagnosis of the 12-lead ECG using a deep neural network." Nature communications 11.1 (2020): 1760.
>
> [3] Natarajan, Annamalai, et al. "A wide and deep transformer neural network for 12-lead ECG classification." _2020 Computing in Cardiology_. IEEE, 2020.
>
> [4] Ogier du Terrail, Jean, et al. "Flamby: Datasets and benchmarks for cross-silo federated learning in realistic healthcare settings." _Advances in Neural Information Processing Systems_ 35 (2022): 5315-5334.
>
> [5] Some of these possibilities could be: Leclerc, Sarah, et al. "Deep learning for segmentation using an open large-scale dataset in 2D echocardiography." IEEE transactions on medical imaging 38.9 (2019): 2198-2210.
>
> [6] Hatamizadeh, Ali, et al. "Unetr: Transformers for 3d medical image segmentation." Proceedings of the IEEE/CVF winter conference on applications of computer vision. 2022.
>
> [7] Tan, Shawn, et al. "Icentia11k: An unsupervised representation learning dataset for arrhythmia subtype discovery." arXiv preprint arXiv:1910.09570 (2019).

---

### Official Review · Reviewer_19DC · 2024-11-05

**Soundness:** 3
**Presentation:** 3
**Contribution:** 3
**Rating:** 6
**Confidence:** 4

**Summary:**

In this paper, the authors present a benchmark for federated learning applied to two cardiovascular disease related modalities: ECG and ECHO. The ECG dataset comprises 4 publicly available datasets, while the ECHO dataset comprises 3 publicly available datasets. The authors evaluate several existing SOTA approaches, including FedAvg, FedProx, Scaffold, FedInit, Ditto, FedSM, and FedALA, and investigate several settings: local vs global test set performance, long-tail challenge, non-IID challenge, and label incompleteness.

**Strengths:**

- The paper aggregates several publicly available datasets that are useful for advancing research in this area in future work.
- The repository is well organized and easy to navigate.
- The analysis is thorough and spans three different challenges related to FL (long-tail distribution, label incompleteness, and non-IID data).
- The authors include several SOTA baselines and describe their hyperparameter tuning strategy.
- They also consistently provide confidence intervals across all results tables.
- The figures and tables are neat and easy to understand.

**Weaknesses:**

- Can you clarify the key differences and potential impact of your benchmark compared to existing work like Goto et al. 2022. Can you also discuss how the focus on long-tailedness and incomplete labels advances the field beyond prior benchmarks?
- To improve the paper's accuracy, can you change the title so that it better reflects the scope, e.g. "FedCVD: A Federated Learning Benchmark for ECG Classification and Echocardiogram Segmentation"?
- The paper requires significant proofreading, i.e. inconsistencies with abbreviations / capitalization / spelling mistakes.
- The abstract should make it clear that the benchmark is using publicly available datasets, rather than presenting new datasets.
- The API functions in Figure 1 are more like functions in their repository so I'm not sure if the term API is the best fit description.
- The dataset sizes and characteristics should be summarized in the main text.
- To better interpret the results, could you discuss why certain FL methods performed better on the long-tail challenge or label incompleteness scenarios?

**Questions:**

Please see above

---

> ### Author Response · Authors · 2024-11-27
> **Responses to Reviewer 19DC (1/2)**
>
> Thank you for your positive and thorough review. We are open to discuss with the reviewer regarding these points as well as the revised paper.
>
> ## Q1: Can you clarify the key differences and potential impact of your benchmark compared to existing work like Goto et al. 2022. Can you also discuss how the focus on long-tailedness and incomplete labels advances the field beyond prior benchmarks?
>
> Thanks for the insightful question. Below, we outline the key distinctions and potential contributions of our benchmark compared to the work by Goto et al. (2022), as well as how our emphasis on long-tail distribution and label incompleteness contributes to advancing the field.
>
> |            | FedHCD                              | FedCVD                                                               |
> | ---------- | ----------------------------------- | -------------------------------------------------------------------- |
> | Algorithms | 1 (FedAvg)                          | 9(FedAvg, FedProx, Scaffold, FedInit, ...)                           |
> | Tasks      | Binary Classification               | Multi-label Classification & Segmentation                            |
> | Metrics    | AUROC                               | Micro-F1, Macro-F1, mean average precision, DICE, Hausdorff distance |
> | Challenges | Non-IID(demographics heterogeneity) | Non-IID, Long-tail distribution, Label incompleteness                |
> | Datasets   | 3 private datasets                  | 7 public datasets                                                    |
>
> As summarized in the table, Goto et al.’s work primarily applies a single federated learning (FL) algorithm, FedAvg, to a binary classification task focused on hypertrophic cardiomyopathy detection. This limits its scope to exploring only a narrow aspect of FL in the context of cardiovascular disease (CVD). In contrast, our benchmark investigates a broader range of FL algorithms (***nine*** in total, including FedAvg, FedProx, Scaffold, and so on) across more diverse and complex tasks, such as multi-label classification and 2D segmentation. Furthermore, while Goto et al.’s work considers Non-IID challenges stemming from demographic heterogeneity, our benchmark incorporates additional, highly relevant challenges in FL scenarios, namely long-tail distribution and label incompleteness.
>
> **Addressing these two challenges is particularly crucial in real-world FL applications**, including healthcare. Long-tail distribution arises naturally in fine-grained medical tasks, where some institutions may only possess a small number of rare disease samples. Existing FL methods often fail to preserve knowledge from such rare but critical classes, leading to suboptimal performance. By contrast, our benchmark highlights this issue, enabling the development and evaluation of algorithms that better capture these underrepresented cases. Similarly, label incompleteness is prevalent in federated healthcare due to varying institutional priorities and capacities, where different institutions annotate overlapping but non-identical subsets of labels. Designing methods to effectively leverage partially annotated data can significantly enhance model generalization and utility across all participating parties.
>
> To the best of our knowledge, prior benchmarks, including Goto et al., have not thoroughly considered these challenges, which are critical for improving the robustness and applicability of FL in real-world medical contexts. By incorporating long-tail distribution and label incompleteness into our benchmark, we aim to provide the community with a more rigorous and realistic framework for evaluating FL approaches in healthcare, paving the way for future innovations.
>
> ## Q2: To improve the paper's accuracy, can you change the title so that it better reflects the scope, e.g. "FedCVD: A Federated Learning Benchmark for ECG Classification and Echocardiogram Segmentation"?
>
> We appreciate the reviewer’s thoughtful suggestion regarding the title. To better align with the scope of our work, we have revised the title in the updated submission to:**“FedCVD: A Federated Learning Benchmark for ECG Classification and Echocardiogram Segmentation”**. This updated title more accurately reflects the focus and contributions of our benchmark. Thank you for helping us enhance the clarity and precision of the manuscript.
>
> ## Q3: The paper requires significant proofreading, i.e. inconsistencies with abbreviations / capitalization / spelling mistakes.
>
> Thank you for pointing this out. We have thoroughly reviewed the manuscript to address inconsistencies in abbreviations, capitalization, and spelling. Specifically, we ensured consistent usage of terms such as “Fed-ECG” and “Top-K” throughout the text. These updates improve the overall clarity and professionalism of the manuscript. We appreciate your feedback in helping us refine our submission.

---

> ### Author Response · Authors · 2024-11-27
> **Responses to Reviewer 19DC (2/2)**
>
> ## Q4: The abstract should make it clear that the benchmark is using publicly available datasets, rather than presenting new datasets.
>
> We appreciate the reviewer’s suggestion and have updated the abstract ***(Line 27-28)*** to clarify that our benchmark utilizes publicly available datasets rather than introducing new ones.
>
> ## Q5: The API functions in Figure 1 are more like functions in their repository so I'm not sure if the term API is the best fit description.
>
> Thank you for your observation. After considering your suggestion, we have replaced the term “API” with “Framework” in the manuscript for greater accuracy ***(Line 65-70)***. Additionally, Figure 1 has been updated to reflect this change. We appreciate your feedback, which has helped us improve the precision of our terminology and presentation.
>
> ## Q6: The dataset sizes and characteristics should be summarized in the main text.
>
> Thank you for the suggestion. We have updated the manuscript to include a detailed summary of dataset sizes and characteristics, presented in Table 2 ***(Line 216-225)***. This addition provides readers with a clear understanding of the scope and diversity of the datasets used in our benchmark. We appreciate your feedback, which has helped us enhance the completeness and clarity of our work.
>
> ## Q7: To better interpret the results, could you discuss why certain FL methods performed better on the long-tail challenge or label incompleteness scenarios?
>
> Thank you for your insightful suggestion.
>
> Intuitively, FL methods designed to address non-IID challenges tend to perform better in scenarios involving long-tail distributions and label incompleteness. In our ECG dataset, for example, different institutions exhibit varying intra-institution long-tail distributions. Algorithms such as FedProx and Scaffold leverage regularization techniques and control of variables, effectively utilizing the diverse training samples from different institutions. This helps mitigate the adverse impact of long-tail distributions.
>
> Label incompleteness, which is another form of non-IID data, presents challenges similar to long-tail distributions. FL methods like FedProx and Scaffold are still more effective at handling such scenarios compared to FedAvg. These methods better control non-IID situations, allowing them to make more effective use of partially annotated data across institutions.

---

> > ### Comment · Reviewer_19DC · 2024-11-28
> >
> > Thank you to the authors for providing responses to my questions. While they have been addressed, I will maintain my score as I am not sure whether ICLR would be the best venue. If the other reviewers and program chair believe that it is, then I am happy for it to be accepted.

---

### Meta-Review · Area_Chair_4tdv · 2024-12-21

**Metareview:**

The paper introduces a benchmark for federated learning for cardiovascular disease data, including ECG classification and echo segmentation. The benchmark uses several already-available public datasets and analyzes long-tail behavior and label incompleteness.


Strengths:
 - The benchmark uses public datasets to demonstrate the performance of federated learning on common data challenges in CVD settings
 - The authors were very responsive during the discussion. They actively performed new experiments and updated confusing sections of the manuscript.

Weaknesses
 - The largest weakness is that it is unclear how this fits with the ICLR conference scope. The specificity of the paper on one narrow set of FL algorithms on two CVD tasks means that it is not clear how much the larger ICLR community will benefit. For example, if researchers are similarly interested in non-IID labels in federated learning settings, it is not obvious whether this paper has generalizable insights.
 - Additionally, reviewers consistently raised concerns about clarity issues in the work. The lack of detailed analysis about comparisons to prior work (e.g., Goto et al, 2022) was brought up several times.
 - There were also several issues about the definitions of variability, the SSL experiment setup, the limited number of architectures examined, and the lack of details in the framework of the experiment setup.


I applaud the authors for extensive responses to reviewer concerns, including entirely new Transformer-based architecture experiments. However, the fit with ICLR and the extent of modifications needed weighed heavily in my decision.

**Additional Comments On Reviewer Discussion:**

In response to reviewers, the authors made significant changes to the paper throughout the discussion process. Modifications included clarifying the relationship with prior work (i.e., Goto et al, 2022). Several writing issues in the paper (e.g., spelling mistakes, abbreviations, capitalizations) were raised and addressed.  An additional Transformer-based model was added. The authors also added label-specific performance metrics.

I will note that the reviewers did not respond to the extensive paper edits; however, reviewers were not expected to respond to all modifications, especially on a paper like this one that required so many changes. Notably, several of the author responses, e.g., variability, felt incomplete since five random seeds is insufficient to determine variability of the experiments. Another round of reviewer responses to the paper modifications is beyond the scope for one ICLR submission, and therefore the paper must be judged in its current form.

---

### Decision · Program_Chairs · 2025-01-22

Reject